# Neural expressiveness for beyond importance model compression

## Abstract

Neural Network Pruning has been established as driving force in the exploration of memory and energy efficient solutions with high throughput both during training and at test time. In this paper, we introduce a novel criterion for model compression, named "Expressiveness". Unlike existing pruning methods that rely on the inherent "Importance" of neurons' and filters' weights, "Expressiveness" emphasizes a neuron's or group of neurons ability to redistribute informational resources effectively, based on the overlap of activations. This characteristic is strongly correlated to a network's initialization state, establishing criterion autonomy from the learning state (*stateless*) and thus setting a new fundamental basis for the expansion of compression strategies in regards to the "When to Prune" question. We show that expressiveness is effectively approximated with arbitrary data or limited dataset's representative samples, making ground for the exploration of *Data-Agnostic strategies*. Our work also facilitates a "hybrid" formulation of expressiveness and importance-based pruning strategies, illustrating their complementary benefits and delivering up to $10\times$ extra gains w.r.t. weight-based approaches in parameter compression ratios, with an average of 1% in performance degradation. We also show that employing expressiveness (independently) for pruning leads to an improvement over top-performing and foundational methods in terms of compression efficiency. Finally, on YOLOv8, we achieve a 46.1% MACs reduction by removing 55.4% of the parameters, with an increase of 3% in the mean Absolute Precision ($mAP_{50-95}$) for object detection on COCO dataset.

## 1 Introduction

To address the computational constraints of existing models, Model Compression [7] has emerged as a prominent solution in exploring models that achieve comparable performance, but with reduced computational complexity [52]. Within this scope, Floating Point Operations (*FLOPs*) are used to estimate a model's computational complexity, by measuring the arithmetic operations required for a forward pass, while parameters (*params*) are associated with a model's size in terms of memory space [48] and their reduction can be a precursor towards more energy efficient solutions [5]. Although *FLOPs* and *params* often correlate, their relationship isn't strictly linear. For instance, VGG16 [43] has $17\times$ more parameters than ResNet-56 [17] but only $3\times$ more *FLOPs*, largely due to VGG16's extensive use of fully connected layers. At first sight, this can be attributed to the differences in network topologies. From a deeper perspective, the intricacies of various operations at handling computational workloads, such as residual structures [17, 55], depthwise separable convolutions [19], inverted residual modules [18], channel shuffle operations [59] and shift operations [53], coupled with their interplay, may significantly affect the relationship between *FLOPs* and *params* in a neural network. In a nutshell, besides the use of more computationally efficient operations as above-

mentioned, Model Compression aims to maintain model performance while optimizing the two aforementioned metrics via tensor decomposition, data quantization, and network sparsification [7].

In this paper we emphasize on the sparsification strategy of pruning [49], which we use as a basis framework to introduce **"Expressiveness"** as a new criterion for compressing neural networks. Existing pruning methods focus on removing redundant network elements – be they weights, neurons, or structures of neurons – in ways that minimally affect the overall performance of a network, based on the criterion of **"Importance"**, e.g. [38, 58, 20, 30]. Importance-based methods address questions like "How much does the removal of a network's element cost in terms of performance degradation?" and "How much information does a network element contain?" in various ways. More specifically, they are motivated by the information inherent in network elements, such as the magnitude of weights [15, 28], similarity of weights or weight matrices [29, 60] ; and their sensitivity to the network's loss function, such as the magnitude of gradients [38] and more [49, 3]. Such dependencies on weights' distributions constitute the aforementioned pruning methods to be "data-aware" since they intrinsically rely on the input data and the information state of the model, making the importance estimation of the network's elements challenging and often costly due to factors like i) the stochasticity from training with minibatches, ii) the presence of plateau areas in the optimization space, and iii) the complexity introduced by nonlinearities [38]. Liu et al. [36] have also discussed limitations in the perception of importance within trained models, i.e. the authors criticize the ability of network's elements importance to generalize to pruned derivatives, while also questioning the necessity of training large-scale models prior pruning.

Inspired by the concepts of "Information Plasticity" [2] and the "Lottery Ticket Hypothesis" (LTH) [12], we aim to address the limitations of previous importance-based methods through elaborating the "Expressiveness" criterion in model compression. In contrast to "Importance", we focus on understanding the capability of network elements to redistribute informational resources to subsequent network elements. We define "Expressiveness" as - "*A neuron's or group's of neurons potential (when a network is not fully trained) or ability (when it is trained) to extract features that maximally separate different samples*". As derived by [2], the early training phase of a model is crucial in shaping its expressiveness, with the formation of critical paths —strong connections that determine the "workload distribution"— being particularly significant during these initial stages. It's essential to note that the network's initialization state influences the formation of those paths, which interestingly enables "Expressiveness" to be a fit criterion for compression during all time instances of a networks' convergence [12], setting a baseline for answering the question of "When to prune?" [42]. *Our proposed pruning metric centers on measuring the overlap of activations between datapoints of the feature space*. In that way, expressiveness is based on effectively evaluating the inherent ability of the network's neurons to differentiate sub-spaces within the feature space. We experimentally show that utilizing either small sets of arbitrary data points from the feature space or stratified sampling [34] from each class yields consistent estimations of expressiveness. Finally, we propose and implement a new "hybrid" pruning optimization strategy that cooperatively searches, exploits and characterizes the complementary benefits between "Importance" and "Expressiveness" for model compression. In summary, this work offers the following four-fold contribution: (i) we propose Expressiveness, a novel criterion based on the overlap of activations for model compression; (ii) we provide an in-depth theoretical analysis of both the fundamental principles and the technical intricacies of the proposed criterion; (iii) we validate the hypothesis that Expressiveness can be approximated with little to none input data, opening the road for data-agnostic pruning strategies; and (iv) through extensive experimentation we offer a thorough comparison w.r.t to both foundational and state-of-the-art methods demonstrating the efficiency and effectiveness of the proposed technique in model compression, while also examining the feasibility and effectiveness of a "hybrid" expressiveness-importance pruning strategy.

Specifically, we validate "Expressiveness" on the CIFAR-10 [24] and ImageNet [40] datasets using a variety of models with different design characteristics [44, 17, 45, 21, 19]. We demonstrate the superiority of our novel criterion over existing solutions, including many top performing structural pruning methods [31, 61, 58, 32, 23, 46, 11], and show significant params reduction while maintaining comparable performance. We experimentally explore and analyze the complementary nature of expressiveness and importance, showing that summary numeric evaluation provides up to 10× additional parameter compression ratio gains, with an average of 1% loss decrease w.r.t group $\ell$1-norm [28]. Finally, we experiment on the current state-of-the-art computer vision model (YOLOv8 [9, 22]), showcasing notable compression rates of 53.9% together with performance gains of 3% on

the COCO dataset [33], and highlighting the ability of more expressive neurons to better recover lost information from the pruning operation.

## 2  Related Work

**Weight (Non-Structural) Importance.** Han et al. [15, 14] and Guo et al. [13] approached the importance of weights based on their magnitude, removing connections below given thresholds. However, earlier works [25, 16] emphasized on the Hessian of the loss and have questioned whether magnitude is a reliable indicator of weight's importance, as small weights can be necessary for low error. In this direction, several studies [4, 47, 41, 8] have proposed strategies of iterative magnitude pruning, in the form of "adaptive weight importance", where weights are ranked based on their sensitivity to the loss. From a different perspective, Yang et al. [56] address the limitations of individual weight's saliency that fail to account for their collective influence and provide a formulation of weight's importance based on the error minimization of the output feature maps. Expanding on this concept, Xu et al. [54] propose a layer-adaptive pruning scheme that encapsulates the intra-relation of weights between layers, focusing on minimizing the output distortion of the network. Amongst other factors and limitations (as also discussed in 1), weight importance is very expensive to measure, mainly because of the increased complexity induced by the mutual influences of the weights among interconnected neurons. This, coupled with the requirement for specialized hardware to manage the irregular sparsity patterns resulting from weight pruning [57], has shifted research focus towards structural pruning [28], where neurons or entire filters are removed.

**Neuron and Filter (Structural) Importance.** Many where driven by the success of Iterative Shrinkage and Thresholding Algorithms (ISTA) [6] in non-structural sparse pruning and proposed filter-level adaptations [28, 29, 32, 26], based on the relaxation ($\ell 1$ and $\ell 2$) of $\ell 0$ norm minimization. However, the loss of universality of such magnitude-based methods remains a limitation in the approximation of importance even in the structural scope. Yu et al. [58] further elaborate on the idea of error propagation ignorance, where the analysis is limited to the statistical properties of a single [28, 29] or two consecutive layers [37]. The authors suggest that the importance of neurons is better approximated from the minimization of the reconstruction error in the final response layer from which it is propagated to previous layers. In contrast to this view, Zhuang et al. [61] emphasize on the discriminative power of a filter as a more effective measure of importance and highlight that this aspect is not effectively assessed by the minimization of the reconstruction error. In a manner that reflects the progression of weight importance, Molchanov et al. [38] define "adaptive filter importance" as the squared change in loss and apply first and second-order Taylor expansions to accelerate importance's computations. Predominantly, the data-awareness imposed by most pruning strategies is added to their already high-complexity – i.e. mostly non-convex, NP-Hard problems that require combinatorial searches. This renders the estimation of importance both computationally expensive and labor-intensive, similarly to non-structural approaches. Notably, Lin et al. [30] propose a less data-dependent solution based on the observation that the average rank of multiple feature maps generated by a single filter remains constant. HRank [30], alongside several other feature-guided filter pruning approaches, are valuable indicators towards data independence. Such works form a principle that pruning elements are better evaluated in the activation phase, where the importance of information and the richness of characteristics for both input data and filters are better reflected. In this work, we expand on this belief and we through extensive experimental analysis, we demonstrate that neither the information state nor the input data is required for the discriminative characterization of an element.

## 3  Neural Expressiveness

### 3.1  Weights and Activations: Importance vs Expressiveness

Neurons are the main constituent element of a neural network. Given a neural network $\mathcal{N}$, we denote neurons by $a_i^{(l)}$, where $l \in L$ is indicative of the neuron's layer in a network with $L = \{l_0, ..., l_l, ..., l_{|L|}\}$ layers and $i$ of its position in the given layer $l = \{a_0, ..., a_i, ..., a_{|l|}\}$. Another important element are the learning parameters of the network. Otherwise the weights represent the strength of connections between neurons in adjacent layers and are denoted by $w_{ij}^{(l)}$, where $i$ and $j$ index the neurons in the current and previous layers. In that manner, neuron's can be perceived as

switches that allow or block information from propagating through-out a network. The activation (or not) of a neuron $a_i^{(l)}$ depends on the output value of its activation function $\sigma(\cdot)$, where there are many popular options for the definition of $\sigma$, e.g., sigmoid, tanh, and ReLU functions. Specifically, a neuron's output is defined as follows,

$$a_i^{(l)} = \sigma\left(\sum_j w_{ij}^{(l)} a_j^{(l-1)} + b_i^{(l)}\right) \tag{1}$$

where $b_i^{(l)}$ denotes the bias term. From eq. 1, we observe that a neuron's activation is affected by the activation of the previous layers, hence affecting in the same way the consecutive layers. This interdependence between activations $a^{(l)}$, for a given layer $l$ defines a recurrent form that can be generalized as follows,

$$a^{(l)} = \sigma\left(W^{(l)} f\left(a^{(l-2)}, \ldots, a^{(1)}\right) + b^{(l)}\right). \tag{2}$$

On the other hand, weights are a more static representation of information as they modulate how much influence one neuron's activation has on another's, compared to activations that control the flow of information in a network. This differentiation has motivated us to define two axes of study in the categorisation of pruning criteria, one based on the weights ("importance") and one based on the activation phase ("expressiveness").

**Generalization of concepts in a structural level.** The aforementioned principles extend to the structural representations of weights and activations, the most common being Convolutional Neural Networks (CNNs). For a CNN model with a set of $K$ convolutional layers, where $C^l$ is the $l-th$ convolutional layer. We denote filters (weight maps) and feature maps (activation maps) as $F_k^l$ and $C_k^l$ respectively, where $k$ the is index within a layer. Given filter with dimensions $m \times n$, eq. 1 is adapted as follows,

$$C_k^{(l)}(x,y) = \sigma\left(\sum_{i=1}^m \sum_{j=1}^n F_{ij}^{(l,k)} a_{x+i-1,y+j-1}^{(l-1)} + b_k^{(l)}\right) \tag{3}$$

where $(i,j)$ and $(x,y)$ are the coordinates of weights and output activations within the filter and the output activation map respectively. Similarly, a convolution layer $l$ can be analyticaly expressed as follows,

$$C^{(l)} = \begin{cases} \sigma\left(\bigoplus_{k=1}^{K^{(1)}} F^{(1,k)} * X + B^{(1)}\right) & \text{if } l = 1 \\ \sigma\left(\bigoplus_{k=1}^{K^{(l)}} F^{(l,k)} * C^{(l-1)} + B^{(l)}\right) & \text{if } l > 1 \end{cases} \tag{4}$$

with $X$ being the input to the first layer of the network, and where symbol $*$ denotes convolution operation and $\bigoplus$ denotes the concatenation operation. Within this context[1], eq. 2 is generalized as follows,

$$C^{(l)} = \sigma\left(\bigoplus_{k=1}^{K^{(l)}} F^{(l,k)} * f\left(C^{(l-2)}, \ldots, C^{(1)}\right) + B^{(l)}\right). \tag{5}$$

**Conceptualization of information propagation.** Consider a task with $X = \{x_i\}_{i=1}^{|D|}$ denoting dataset samples, where $|D|$ is the size of the dataset. Given the information state (weight state) of a CNN model with $K$ convolutional layers at a given time $t_i$, X is mapped through the network as $f(X, W_{t_i})$, where $W_{t_i} = \{F_{t_i}^1, \ldots, F_{t_i}^l, \ldots, F_{t_i}^{|K|}\}$ and $F_{t_i}^l = \{F_{t_i}^{(l,1)}, \ldots, F_{t_i}^{(l,k)}, \ldots, F_{t_i}^{(l,K^{(l)})}\}$, with $K^{(l)}$ being the amount of weight maps (filters) in a given layer $l$. This process can be further analyzed as follows,

$$f(X, \mathbf{W}_{t_i}) = \mathcal{F}_{|K|}(\mathcal{F}_{|K|-1}(\ldots \mathcal{F}_1(X; \mathbf{F}_{t_i}^1); \mathbf{F}_{t_i}^2); \ldots; \mathbf{F}_{t_i}^{|K|}), \tag{6}$$

where $\mathcal{F}_l$ represents the mapping operation of convolutional layer $l$.

Based on eq. 2 and eq. 5, the equivalent of the previous based on the activations of the layers can be expressed as,

$$f(X, \mathbf{W}_{t_i}) = C^{(|K|)}\left(\ldots \left(C^{(2)}\left(C^{(1)}\left(X, \mathbf{F}_{t_i}^1\right), \mathbf{F}_{t_i}^2\right) \ldots\right), \mathbf{F}_{t_i}^{|K|}\right). \tag{7}$$

---

[1]We do not include pooling and batch normalization layers in the formulations; however, the equations can be expanded to incorporate them as intermediate steps based on each architecture.

Here, $C^{(l)}$ represents the activation map of the $l$-th layer, where $C^{(l)} = \mathcal{F}_l(C^{(l-1)}; \mathbf{F}_{t_i}^l)$ aligns with the structure defined in eq. 4. In this formulation, $C^{(1)}$ is the activation map of the first layer, computed using the input $X$ and the first layer's filters $\mathbf{F}_{t_i}^1$. Subsequent layers' activation maps $C^{(l)}$ are derived from the previous layer's output $C^{(l-1)}$ and their respective filters $\mathbf{F}_{t_i}^l$. Assuming a classification task, the final layer $C^{(|K|)}$ is considered the classification layer, effectively summarizing the hierarchical feature extraction and transformation process across all convolutional layers.

## 3.2 Mathematical Foundation of Neural Expressiveness.

We observe that the training parameters of the model, in this case $W_{t_i}^2$, are responsible for transforming the original input feature space $X$ into a sequence of intermediate feature spaces $\{C^{(1)}, \ldots, C^{(|K|-1)}\}$, progressing towards the final prediction formulated by the prediction layer $C^{(|K|)}$.

Based on this intrinsic characteristic of neural networks and inspired by the goal of optimizing feature discrimination, akin to the entropy reduction strategy in decision trees [51], we assess network elements ability, in this scenario filters, to extract features, i.e., activation patterns, that maximally separate different input samples $x_i$. In other words, we score the expressiveness of the filters within $W_{t_i}$, based on the discriminative quality of the intermediate feature spaces they generate, where the feature space generated by a filter $F_k^l$, is denoted as $C_k^l$.

**Neural Expressiveness foundational concept.** When assessing the expressiveness of an element within $W_{t_i}$ based on its generated feature spaces, e.g., $NEXP(F_{t_i}^l; C^l)$, we cooperatively evaluate all of its preceding elements, as derived from eq. 5. This can be formulated as,

$$NEXP(F_{t_i}^l; C^l) = NEXP(F_{t_i}^l; (C^{(l-1)}, C^{(l-2)}, \ldots, C^{(1)})), \tag{8}$$

which can be further extended to incorporate the inter-dependencies between the examined element and its predecessors, in accordance with eq. 7, as detailed below:

$$NEXP\left(F_{t_i}^l; \left(C^{(l-1)}, C^{(l-2)}, \ldots, C^{(1)}\right)\right) =$$
$$NEXP\left(F_{t_i}^l; \left((C^{(l-2)}, \mathbf{F}_{t_i}^{l-1}), (C^{(l-3)}, \mathbf{F}_{t_i}^{l-2}), \ldots, (X, \mathbf{F}_{t_i}^1)\right)\right). \tag{9}$$

The aforementioned eqs. 8 and 9 provide the *foundational concepts for utilizing the evaluation of the activation phase*, in an endeavor to encourage the development of more universal solutions by addressing the limitations of universality inherent in the assessment of the weight state alone (as also discussed in sections 1 and 2).

**Formulation of Neural Expressiveness (NEXP) Score.** Diving deeper into the Neural Expressiveness (NEXP) scoring process, we follow eq. 9 previously and assume a mini-batch $X^{'} = \{x_i^{'}\}_{i=1}^N$, with $N$ being the number of samples in it. Mapping the batch through the network, based on eqs. 6 and 7, generates a set of sequences of feature spaces (activation maps), denoted as $S = \{s_1, \ldots, s_i, \ldots, s_N\}$, where $s_i = \{x_i^{'}, \ldots, C_i^l, \ldots, C_i^{|K|}\}$ is the sequence of the activation patterns generated from sample $x_i^{'} \in X^{'}$ and $|s_i| = |K| + 1$ is its cardinality, including the feature space of sample $x_i^{'}$. To evaluate a specific filter $k$ in layer $l$, denoted as $F_k^l$, we utilize the retrieved activation patterns from that filter, denoted as $\{s_{i,k}^l\}_{i=1}^N$, where $s_{i,k}^l = C_{i,k}^l$ is the activation pattern retrieved from filter $k$ in layer $l$.

To score the Neural Expressiveness of $F_k^l$, we first construct a $N \times N$ matrix that expresses all possible combinations of the activation patterns derived from the different input samples. This table can be visualised as follows,

$$\begin{pmatrix} s_{(1,1),k}^l & s_{(1,2),k}^l & \cdots & s_{(1,N),k}^l \\ s_{(2,1),k}^l & s_{(2,2),k}^l & \cdots & s_{(2,N),k}^l \\ \vdots & \vdots & \ddots & \vdots \\ s_{(N,1),k}^l & s_{(N,2),k}^l & \cdots & s_{(N,N),k}^l \end{pmatrix}. \tag{10}$$

---

[2]Bias terms are excluded for simplicity.

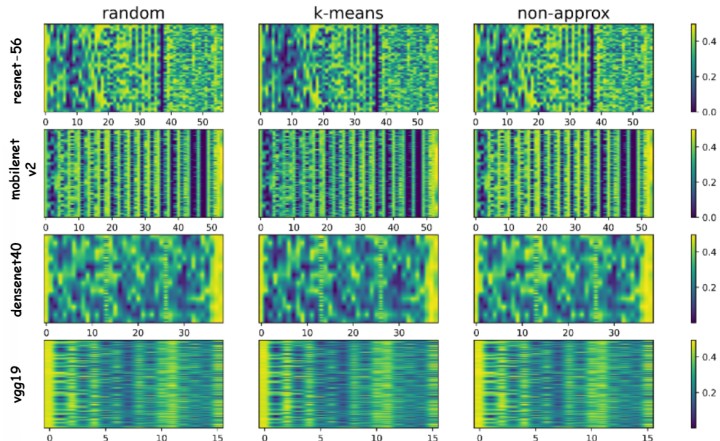

Figure 1: **Expressiveness statistics of feature maps from different convolutional layers and architectures on CIFAR-10.**

where $s^l_{(i,j),k}$ denotes the dissimilarity of activations patterns between the $i$-th and the $j$-th sample of the batch. In other words, the matrix in eq. 10 represents all the possible combinations of NEXP calculations, where each element $s^l_{(i,j),k}$ derives from $f(s^l_{i,k}, s^l_{j,k})$, with $f$ being any dissimilarity function. Without loss of generality, for the rest of the study, we use the Hamming distance as the operator implementing dissimilarity function. Activations are first binarized (values greater than 0 become 1, and the rest become 0), i.e. enabling to evaluate the degree of overlap between the binary activation patterns using $f$.

We note that the matrix's diagonal, where $i$ equals $j$, along with the elements below the diagonal, where $i$ is greater than $j$, do not contribute additional value to quantifying the discriminative ability of an element. The diagonal elements represent comparisons of the same sample's activation patterns, rendering them redundant. Meanwhile, the lower triangular elements are considered duplicates since $s^l_{(i,j),k}$ is equal to $s^l_{(j,i),k}$, thereby not adding any new information. Drawing from these two observations, we define the Neural Expressiveness score (NEXP) as follows,

$$NEXP(F^l_k) = \frac{1}{\frac{N(N-1)}{2}} \sum_{i=1}^{N} \sum_{j=i+1}^{N} f(s^l_{i,k}, s^l_{j,k}) \tag{11}$$

The **more similar** the activation patterns derived from an element are, the **less expressive** it is declared to be. In eq. 11, we also normalize the score w.r.t the total amount of combinations ($\frac{N(N-1)}{2}$), thereby deriving the average expressiveness score. This average score is then utilized to characterize the discriminative capability/capacity of the examined network element. In this study, we used the mean operation, however, we note that alternate statistical measures, e.g., minimum, maximum, median, etc., could feasibly be applied in the computation of the overall score.

### 3.3 Dependency to Input Data

NEXP evaluates the inherent property of network elements to maximally distinguish between input samples. We extend this line of thought and assess its sensitivity to input data $X$ and mini-batch size $N$, in order to delineate the dependence between NEXP and the input data. To achieve that, we perform a sensitivity analysis of NEXP to the mini-batch data $X$, using two input sampling strategies to assemble a batch with 60 samples, namely random sampling (denoted as 'random') and class-representative sampling via k-means (denoted as 'k-means'). We define the true NEXP score (denoted as 'non-approx') for each filter as the value obtained by comparing all activation patterns across the *entire training dataset* (more info in A.1). Fig. 1 presents a detailed comparative illustration of the results that highlight the similarities in NEXP estimations across various trained networks, including VGGNet [44], ResNet [17], MobileNet [19] and DenseNet [21] on CIFAR-10 dataset. Columns represent the aforementioned sampling strategies, while colors indicate expressiveness levels, with higher values signifying greater expressiveness. In each sub-figure, the x-axis indicates

**Algorithm 1** NEXP Pruning Algorithm

**Define:** $NEXP_{\text{map}} = \{\{NEXP(F^l_k)\}^{|C^l|}_{k=1}\}^{|K|}_{l=1}$

**Require:** A mini-batch $X$, a neural network $\mathcal{N}(W_{t_i})$, a theoretical speed-up target, denoted $\tau$, and the allowed amount of pruning steps, denoted $\text{steps}_{\text{max}}$.

**Ensure:** $\frac{\text{FLOPs}(\mathcal{N})}{\text{FLOPs}(\mathcal{N}_{\text{pruned}})} \geq \tau$

1: Initialize $NEXP_{\text{map}} \leftarrow f(X; W_{t_i})$
2: Initialize $\tau_{\text{current}}$ as 1
3: Initialize $\text{steps}_{\text{current}}$ as 1
4: Initialize $\mathcal{N}_{\text{pruned}}$ as $\mathcal{N}$
5: **while** ($\tau_{\text{current}} < \tau$) and ($\text{steps}_{\text{current}} \leq \text{steps}_{\text{max}}$) **do**
6:     $F_{\text{to\_prune}} = bottom_\kappa(NEXP_{map})$
7:     $\mathcal{N}_{\text{pruned}}(W_{\text{pruned}}) = prune(\mathcal{N}_{\text{pruned}}, F_{\text{to\_prune}})$
8:     $NEXP_{\text{map}} \leftarrow f(X; W_{\text{pruned}})$
9:     $\tau_{\text{current}} = \frac{\text{FLOPs}(\mathcal{N})}{\text{FLOPs}(\mathcal{N}_{\text{pruned}})}$
10:     $\text{steps}_{\text{current}} + +$
11: **end while**
12: **return** $\mathcal{N}_{\text{pruned}}(W_{\text{pruned}})$

Figure 2: **Pruning YOLOv8m trained on COCO for Object Detection.** Comparative results between neural expressiveness (NEXP) and layer-adaptive magnitude-based pruning method (LAMP) [26]. More comparisons in the supplementary material.

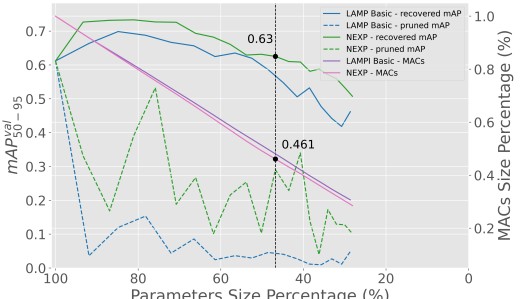

convolutional layer indices, and the y-axis shows feature map indices per layer, standardized through pixel-wise interpolation to align with the layer having the most feature maps. Fig. 1 confirms that NEXP can be effectively estimated using random and limited data samples. Detailed results of this analysis, are presented in Appendix A. The comparative analysis reveals that a mini-batch of 60 samples ($0.4\%$ of $D$ in this case) effectively approximates the NEXP scores calculated from the entire dataset, yielding consistent similarity scores above 99% across most similarity metrics (Table. 3).

### 3.4 Pruning Process

Alg. 1 describes the proposed NEXP-based pruning process, and it has been implemented as extension in the DepGraph pruning framework [11]. A target theoretical speed-up is specified, referred to as the Compression FLOPs Ratio ($\downarrow$) and denoted by $\tau$. This ratio is calculated using the formula $\frac{\text{original FLOPs}}{\text{compressed FLOPs}}$. To achieve this target ratio, the network may undergo pruning in one or several steps, dictated by the intricacies of the pruning criterion and adjusted according to the quantity of elements removed at each step. For example, NEXP benefits from additional steps, since a filter's score is reliant on its preceding elements (Section 3.2), and a more gradual update on the scores allows for improved pruning precision. A more in-depth analysis of Alg. 1 along with more details on the implementation options are presented in Appendix B.

## 4 Experimental Evaluation

Details on the experimental settings can be found in Appendix C, including the (a) Datasets and Models (C.1), (b) Adversaries (C.2), (c) Evaluation Metrics (C.3) and (d) Configurations (C.4).

### 4.1 Comparison w.r.t. State-of-Art Model Compression Strategies

**Image Classification on CIFAR-10 and Imagenet-1k.** We compare against a plethora of foundational and top-performing approaches, ranging from filter magnitude-based [28, 32, 29] and loss sensitivity-based [58] methods to feature-guided strategies [23, 30] and search algorithms [35, 31].

**Outcomes and Discussion.** Our findings for various target FLOPs pruning ratios are presented in Tab. 1 (and Tab.6-9 in Appendix D.2) for CIFAR-10, and in Tab. 2 for ImageNet. It is essential to acknowledge the subjectivity in reported performance metrics (accuracy), influenced by the fine-tuning process post-pruning, e.g. the authors in DCP [61] fine-tune for 400 epochs, in contrast to ours 100. We observe that our approach yields consistent improvements in params reduction compared to other methods for given FLOPs ratios, which notably scale significantly for regimes of higher target FLOPs compression ratios $\tau$. For example, on ResNet-56 we show $+0.92\times$ average params reduction gains in the $2\times$-$2.20\times$ FLOPs reduction regime, with -0.38%, +0.05% and -0.37% percentage difference in loss respectively to ABC [31], SCP [23] and HRank [30], while on ResNet-110 we

Table 1: Analytical Comparison of Importance-based solutions and Expressiveness on CIFAR-10 using ResNet architectures [17] - ResNet-56 (left) and ResNet-110 (right).

| Method | top-1 acc Base (%) | Δ (%) | Compression Ratio ↓ #Params | #FLOPs |
|---|---|---|---|---|
| L1 [28] | 93.06 | +0.02 | 1.16× | 1.37× |
| NEXP (Ours) | 93.36 | +0.05 | **1.69×** | 1.53× |
| GAL-0.6 [32] | 93.26 | +0.12 | 1.13× | 1.60× |
| NISP-56 [58] | - | -0.03 | 1.74× | 1.77× |
| DCP-Adapt [61] | 93.80 | +0.01 | 3.37× | 1.89× |
| HRank [30] | 93.26 | -0.09 | 1.74× | 2.01× |
| SCP [23] | 93.69 | -0.46 | 1.94× | 2.06× |
| NEXP (Ours) | 93.36 | -0.41 | **2.87×** | 2.11× |
| ABC [31] | 93.26 | -0.03 | 2.18× | 2.18× |
| NEXP (Ours) | 93.36 | -1.58 | **4.3×** | 2.50× |
| GAL-0.8 [32] | 93.26 | -1.68 | 2.93× | 2.51× |
| HRank [30] | 93.26 | -2.54 | 3.15× | 3.86× |
| NEXP (Ours) | 93.36 | -5.12 | **21.5×** | 5.00× |

| Method | top-1 acc Base (%) | Δ (%) | Compression Ratio ↓ #Params | #FLOPs |
|---|---|---|---|---|
| L1 [28] | 93.55 | +0.02 | 1.02× | 1.19× |
| NEXP (Ours) | 93.79 | +0.66 | **1.10×** | 1.20× |
| GAL-0.1 [32] | 93.50 | +0.09 | 1.04× | 1.23× |
| HRank [30] | 93.50 | +0.73 | 1.65× | 1.70× |
| NISP-110 [58] | - | -0.18 | 1.76× | 1.78× |
| NEXP (Ours) | 93.79 | +0.18 | 1.78× | 1.80× |
| GAL-0.5 [32] | 93.50 | -0.76 | 1.81× | 1.94× |
| HRank [30] | 93.50 | -0.14 | 2.46× | 2.39× |
| NEXP (Ours) | 93.79 | +0.10 | **2.72×** | 2.42× |
| ABC [31] | 93.50 | +0.08 | 3.09× | 2.87× |
| NEXP (Ours) | 93.79 | -0.37 | **3.81×** | 3.01× |
| HRank [30] | 93.50 | -0.85 | 3.25× | 3.19× |
| NEXP (Ours) | 93.79 | -0.59 | **4.38×** | 3.27× |

show +1.21× average params reduction gains in the 2.87×-3.27× FLOPs reduction regime, with -0.67% and +0.26% percentage difference in loss respectively to ABC [31] and HRank [30]. Similar observations are evident across all tables, where in certain regimes we also show notable performance gains, up to +1.5%, especially for VGGNet, which is more prone to params reductions due to its plain structure.

**Object Detection with YOLOv8.** We evaluate expressiveness against four importance based methods, i.e layer-adaptive magnitude-based pruning (LAMP) [26], network slimming (SLIM) [35], Wang's et al. proposed method (DepGraph) [11] and random pruning that serves as a generic pruning baseline [3]. The experiments were conducted on the YOLOv8m model version [22], utilizing the DepGraph pruning framework [11] with an iterative pruning schedule of 16 steps, where after each pruning step the model was fine-tuned for 10 epochs using the coco128 dataset.

**Outcomes and Discussion.** We report the comparative pruning progress of expressiveness versus the baseline methods, i.e. the remaining percentage of the original model in terms of *MACs* and *params* after each pruning step, named MACs Size Percentage (MSP) and Parameters Size Percentage (PSP) respectively, and highlight the $mAP_{50-95}^{val}$ both after pruning (pruned mAP) and fine-tuning (recovered mAP). We observe that expressiveness outperforms the rest of the reported methods across the whole pruning spectrum, as shown in Fig. 2 (more in Appendix D.2), preserving the initial performance of the model for percentage sizes that reach up to 40% (2.5 ↓) of that of the original model, with less than 0.5% of recovered performance degradation. Our method even achieves a 3% increase in recovered mAP for 46.1% MSP (2.17 ↓), in comparison to the baselines that showcase weak recovery capabilities after the 60% (1.67 ↓) mark in both MSP and PSP. This can be attributed to the intrinsic property of expressiveness to maintain network elements that are more robust to information redistribution, in contrast to "important" labeled structures by other methods. In our experimental scenario, that characteristic is further amplified by the iterative pruning format and the higher amount of fine-tuning epochs at each step, in comparison to conventional pruning schedules that fine-tune for 1 epoch after each iteration or perform a unified fine-tuning session after the last pruning iteration. Interestingly, our criterion also demonstrates significant resistance to performance loss after pruning, achieving 18% increased average performance in terms of pruned mAP compared to the importance-based methods. We have empirically observed that expressiveness benefits from increased cardinality in pruning granularity settings, e.g amount of intermediate steps to achieve a given compression ratio. This stems from expressiveness interactive nature of all elements, as also explained in Sec. 3, where smaller pruning steps combined with iterative fine-tuning, enhance pruning precision and allow for "smoother" redistribution of information in a network, thus contributing to the increased resistance to performance deficits after each pruning step.

## 4.2 Assessing Hybrid Compression space

In this section, we assess the potential efficiency of "hybrid" pruning strategies exploiting the cooperation between importance and expressiveness. We explore the solution space of "hybrid" compression, using a linear combination of importance and neural expressiveness criteria. We guide exploration through the scoring function: $W_{imp} \cdot IMP + W_{nexp} \cdot NEXP$ and conduct experiments with

Table 2: Analytical Comparison of Importance-based solutions and Expressiveness on ImageNet-1k using ResNet-50 [17].

| Method | Base (%) | | Δ Acc (%) | | Compression Ratio | |
|---|---|---|---|---|---|---|
| | top-1 | top-5 | top-1 | top-5 | #Params ↓ | #FLOPs ↓ |
| NISP-50-B [58] | - | - | -0.89 | - | 1.78× | 1.79× |
| NEXP (Ours) | 76.13 | 92.86 | -1.35 | -0.93 | 2.00× | 2.02× |
| ThiNet [37] | 72.88 | 91.14 | -1.87 | -1.12 | 2.06× | 2.25× |
| DCP [61] | 76.01 | 92.93 | -1.06 | -0.56 | 2.06× | 2.25× |
| ABC [31] | 76.01 | 92.96 | -2.49 | -1.45 | 2.27× | 2.30× |
| NEXP (Ours) | 76.13 | 92.86 | -6.77 | -3.43 | **4.05×** | 3.04× |
| GAL-1-joint [32] | 76.15 | 92.87 | -6.84 | -3.75 | 2.50× | 3.68× |
| Hrank [30] | 76.15 | 92.87 | -7.15 | -3.29 | 3.08× | 4.17× |

Figure 3: **Linear exploration of the combinatorial space between importance and expressiveness.**

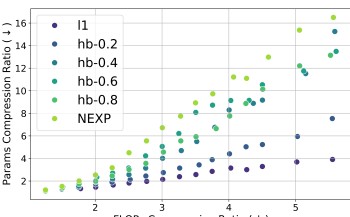

various weight combinations, subject to the constraint $W_{\text{imp}} + W_{\text{nexp}} = 1$. Given that exhaustive search is impractical, we introduce the hyper-parameter $\alpha \in \{0.0, 0.2, \dots, 0.8, 1.0\}$ to restrict the set of permissible combinations, and modify the constraint to $(1 - \alpha) \cdot W_{\text{imp}} + \alpha \cdot W_{\text{nexp}} = 1$. We use group L1-norm [28] as the importance criterion (IMP) and assess all permissible combinations across a linear scale, denoted as $\tau$, representing the target FLOPs compression ratios that we utilized for pruning, on ResNet-56 for CIFAR-10. The outcomes are visualized in Figure 3, which maps our predetermined $\tau$ values on the x-axis against the various parameter compression ratios achieved by each combination. Regarding performance, we report the averaged percentage differences in top-1 accuracy between the baseline importance method (L1) and each hybrid format: -0.21% for hb-0.2, -0.96% for hb-0.4, -1.55% for hb-0.6, -1.07% for hb-0.8, and -2.18% for NEXP.

**Observations.** A consistent pattern is observed across the values of $\alpha$, where larger values yield higher params compression ratios. Notably, hybrid derivatives allow us to explore sub-spaces with higher parameter compression ratios by sacrificing slight performance accuracy. We also observe that the solution vectors corresponding to IMP and EXP act as extremal points in the solution space of hybrid combinations, thus suggesting a degree of partial orthogonality between the two criteria. Furthermore, the findings reveal a polynomial relationship between parameter compression ratios and FLOPs reduction, with compression ratios increasing polynomially to linear increments in FLOPs reduction, and thus enabling more efficient explorations.

### 4.3 Evaluating Neural Expressiveness at Initialization

The nature of NEXP allows to be applied in a weight agnostic manner, i.e. on untrained networks. An extended version of the section's 3.3 analysis, which also includes untrained models (Appendix A), reveals that $NEXP_{\text{map}}$'s obtained at initialization and after network convergence share some expressiveness pattern similarities, particularly in the initial layers. Our numeric evaluation shows a notable correlation between the initialization and converged states for DenseNet-40 and VGG-19, with cosine similarities of 84.10% and 86.82%, respectively. It also indicates greater consistency in neural expressiveness measurements for the first layers of all networks, which could be considered important for the formation of critical paths [2]. Motivated by these observations, we also assess the efficacy of expressiveness as criterion for Pruning at Initialization against various SOTA approaches [27, 50, 46] (Appendix D.1). Our method consistently outperforms (in terms of top-1 acc) all other algorithms, particularly in regimes of lower compression, up to $10^2(\downarrow)$ with an average increase of 1.21% over SynFlow, while maintaining competitiveness at higher compression levels, above $10^2(\downarrow)$ with an average percentage difference of 4.82%, 3.72% and -2.74%, compared to [50], [27] and [46]. In summary, under the assumption that the selection of hyperparameters remains congruent with the initialization [12], consistent map measurements between initial and final states can effectively evaluate NEXP's ability to identify winning tickets. However, a robust evaluation should also consider the initial state quality and the training process, while addressing the "When to prune" question [42].

## 5 Conclusions

In this work, we have introduced "Neural Expressiveness" as a new criterion for model compression. In our NEXP steps, we will explore optimal solutions for the "When" and "How" to prune questions.

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

# A  Duality of Independence: Data $(X)$ and Information State $(W_{t_i})$

Fig. 4 presents a detailed comparative illustration that highlights the similarities in NEXP estimations across various networks, including VGGNet [44], ResNet [17], MobileNet [19] and DenseNet [21] on CIFAR-10 dataset. Specifically, for each network architecture, we showcase expressiveness distributions in both untrained (PaI) and trained (PaT) states. In each sub-figure, the x-axis indicates convolutional layer indices, and the y-axis shows feature map indices per layer, standardized through pixel-wise interpolation to align with the layer having the most feature maps. Columns represent various sampling strategies, while colors indicate expressiveness levels, with higher values signifying greater expressiveness. In other words, the figure illustrates a two-fold sensitivity analysis of NEXP to (i) the mini-batch data ($X$, as outlined in Alg. 1), using two input sampling strategies to assemble a batch with 60 samples, namely random sampling (denoted as 'random') and class-representative sampling via k-means (denoted as 'k-means'), and (ii) the information state ($W_{t_i}$), specifically comparing expressiveness at initialization (PaI) against expressiveness after training (PaT), when weights have converged.

## A.1  True NEXP value (non-approx).

We define the true NEXP score for each filter as the value obtained by comparing all activation patterns across the entire training dataset $D$. In that way, the ability of each element to extract maximal features is evaluated for every data-point in the input feature space of a task at hand. In this study however, due to GPU memory constraints (limited to 12GB of GDDR6 SDRAM), we employed 25% of the total training set, ensuring class distribution is preserved, to determine these exact NEXP scores, denoted as non-approx.

## A.2  Data Agnostic.

To evaluate NEXP's sensitivity to input data, we conduct a similarity analysis for each row in Fig. 4. For each information state (PaI and PaT), we compare the expressiveness map ($NEXP_{\text{map}}$) derived from each sampling strategy against the true NEXP values (non-approx), corresponding to each respective state. For a comprehensive comparison, we utilize various similarity metrics, such as Euclidean Distance, Cosine Similarity, Pearsonr Similarity, and the Structural Similarity Index Measure (ssim_index). Detailed results of this analysis, specific to each state, are presented in Tables 3 (PaT) and 4 (PaI). The comparative analysis reveals that a mini-batch of 60 samples, with a balanced representation from each class, effectively approximates the NEXP scores calculated from the entire dataset, yielding consistent similarity scores above 99% across all similarity metrics for both PaI and PaT. Interestingly, random sampling consistently outperforms the k-means selection strategy, which involves selecting 6 representative samples per CIFAR-10 class. This is especially notable in PaT, with random sampling showing up to a 7.51% higher Pearson correlation, 5% improvement in ssim_index, and 1.14 reduction in Euclidean distance compared to k-means. This further reinforces the statement that comparing activation patterns reflects the intrinsic ability of neural networks to distinguish various input spaces, thus effectively extending the NEXP criterion to random input data and laying the foundation for investigating *Data-Agnostic strategies*.

## A.3  Weight Agnostic

Fig. 4 reveals that $NEXP_{\text{map}}$'s obtained at initialization and after network convergence share some expressiveness pattern similarities, particularly in the initial layers. Detailed comparisons of these similarities across all layers, and specifically for the first five, are presented in Table 5, contrasting the initial maps with the true $NEXP_{\text{map}}$ post-training. The summary of our numeric evaluation confirms a notable correlation between the initialization and converged states for DenseNet-40 and VGG-19, showing up to 84.10% and 86.82% in cosine similarity respectively. It also indicates greater consistency in neural expressiveness measurements for the first layers of all networks, which could be considered important for the formation of critical paths. In this context, the formation of the final state depends on hyperparameter choices, like weight decay and learning rate, and the stochastic nature of training, that could potentially alter the model's progression from its initial state, as also highlighted by Frankle et al. [12]. In that manner, under the assumption that the selection of hyperparameters remains congruent with the initialization, "Expressiveness" can be considered a fit criterion for Pruning at Initialization (PaI). In summary, the consistency of map measurements

between initial and final states may serve as a solid metric for evaluating NEXP's ability to identify winning tickets. Nevertheless, a more robust process of its evaluation should also take into account the quality of the initial state as well as the subsequent training process.

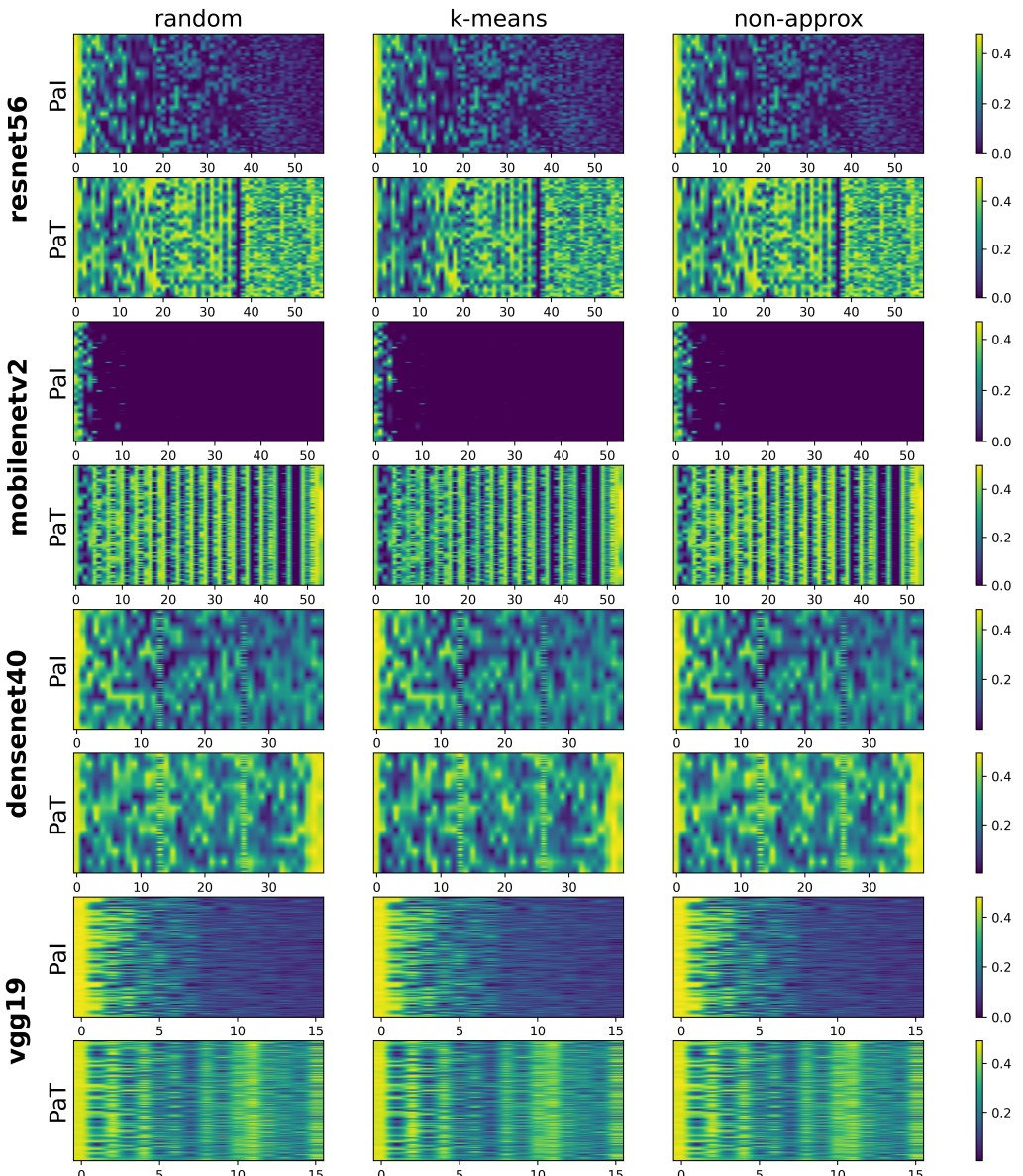

Figure 4: **Expressiveness statistics of feature maps from different convolutional layers and architectures on CIFAR-10 (Extended).** For each architecture we demonstrate the expressiveness distribution for both an untrained instance of the model (PaI), as well as a converged one (PaT). The x-axis represents the indices of convolutional layers and y-axis that of the feature maps in each layer. To maintain consistency across the y-axis, we have interpolated each layer's feature maps (pixel-wise) to match the layer with the most feature maps. Columns denote different sampling strategies and different colors denote different expressiveness values (the higher the value, the more expressive the feature map). To approximate the expressiveness score of each element, denoted as "non-approx", we used 25% of all dataset's samples (not 100% due to memory limitations) maintaining the label's distribution. As can be seen, the rank of each feature map (column of the sub-figure) is almost unchanged (the same color), regardless of the image batches. Hence, even a small number of images can effectively estimate the average rank of each feature map in different architectures.

Table 3: Sensitivity analysis of the input's sampling strategies after training (PaT) using various similarity metrics.

| Model | Sampling Strategy | Euclidean Distance | Cosine Similarity | Pearsonr Similarity | ssim_index |
|---|---|---|---|---|---|
| ResNet-56 [17] | random | **0.2349** | **0.9998** | - | **0.9979** |
| | k-means | 1.3729 | 0.9949 | - | 0.9479 |
| MobileNet-v2 [19] | random | **0.2903** | **0.9994** | **0.9810** | **0.9988** |
| | k-means | 1.1197 | 0.9960 | 0.9059 | 0.9794 |
| DenseNet-40 [21] | random | **0.2751** | **0.9997** | **0.9818** | **0.9970** |
| | k-means | 1.1669 | 0.9959 | 0.9527 | 0.9614 |
| VGG-19 [44] | random | **0.5150** | **0.9989** | **0.9814** | **0.9894** |
| | k-means | 0.8438 | 0.9964 | 0.9556 | 0.9728 |

Table 4: Sensitivity analysis of the input's sampling strategies at Initialization (PaI) using various similarity metrics.

| Model | Sampling Strategy | Euclidean Distance | Cosine Similarity | Pearsonr Similarity | ssim_index |
|---|---|---|---|---|---|
| ResNet-56 [17] | random | **0.1333** | **0.9996** | **0.9979** | **0.9984** |
| | k-means | 0.3948 | 0.9984 | 0.9868 | 0.9859 |
| MobileNet-v2 [19] | random | **0.0340** | **0.9565** | - | **0.9994** |
| | k-means | 0.2441 | 0.9454 | - | 0.9776 |
| DenseNet-40 [21] | random | **0.2297** | **0.9997** | 0.9927 | **0.9977** |
| | k-means | 0.2972 | 0.9994 | **0.9941** | 0.9955 |
| VGG-19 [44] | random | **0.2688** | **0.9988** | 0.9652 | **0.9950** |
| | k-means | 0.4882 | 0.9975 | **0.9724** | 0.9856 |

Table 5: Sensitivity analysis of $NEXP_{\mathrm{map}}$'s retrieved at initialization compared with the true $NEXP_{\mathrm{map}}$ following model convergence.

| Model | Metric | random All | random first-5 | k-means All | k-means first-5 | non-approx (PaI) All | non-approx (PaI) first-5 |
|---|---|---|---|---|---|---|---|
| ResNet-56 [17] | Euclidean Distance | 9.0326 | 5.2005 | **8.8029** | **5.1177** | 8.9986 | 5.1850 |
| | Cosine Similarity | 0.7584 | 0.8765 | **0.7677** | **0.8784** | 0.7592 | 0.8751 |
| | ssim_index | 0.0194 | 0.3794 | **0.0243** | **0.3990** | 0.0206 | 0.3810 |
| MobileNet-v2 [19] | Euclidean Distance | **10.5470** | **7.4966** | 10.6056 | 8.0843 | 10.5492 | 7.5134 |
| | Cosine Similarity | 0.4645 | **0.6478** | 0.4910 | 0.5862 | **0.6702** | 0.6461 |
| | ssim_index | -0.0018 | **0.1187** | -0.0011 | 0.0942 | -0.0020 | 0.1142 |
| DenseNet-40 [21] | Euclidean Distance | 6.1326 | **4.6957** | **6.0594** | 4.7157 | 6.1043 | 4.7364 |
| | Cosine Similarity | 0.8357 | **0.8769** | **0.8410** | 0.8762 | 0.8378 | 0.8761 |
| | ssim_index | **0.0169** | **0.4552** | 0.0101 | 0.4493 | 0.0150 | 0.4464 |
| VGG-19 [44] | Euclidean Distance | 6.3171 | 4.9532 | **6.1194** | **4.8525** | 6.3083 | 4.9810 |
| | Cosine Similarity | 0.8610 | 0.8979 | **0.8682** | **0.9030** | 0.8624 | 0.8972 |
| | ssim_index | 0.0808 | 0.3798 | **0.0812** | **0.3844** | 0.0808 | 0.3712 |

## B Pruning Process: An in-depth analysis

### B.1 Global vs local -scope pruning.

NEXP is used in the pruning process to evaluate and rank different network elements, guiding their subsequent removal based on their scores. In our study, we focused on the removal of filters, i.e., Filter Pruning, where we pruned convolutional structures by removing the least expressive filters. This can be approached in two ways: (i) on a local (layer-by-layer) basis, where filters are assessed and removed according to their expressiveness relative to other filters within the same layer, e.g., eliminating the least $\mu$ expressive filters from each layer. (ii) On a global (network-wide) basis, where all filters across layers are normalized in terms of their scores, allowing for the removal of the least $\kappa$ expressive filters from the entire network. We experimentally observed that "Global Pruning" yields consistent results and outperforms "Local Pruning" when using the NEXP pruning criterion. Therefore, all the experiments reported in this paper were conducted using the "Global Pruning" approach.

### B.2 One-shot vs Iterative pruning.

Furthermore, another design parameter to consider in the pruning process is its coordination with fine-tuning. In this context, two widely adopted strategies are: (a) "One-Shot" pruning, where pruning is completed entirely before any fine-tuning occurs, and (b) "Iterative" pruning, which involves alternating between pruning and fine-tuning via an iterative sequence. The first one (a) can be considered a more lightweight approach and allows for a more robust evaluation of the pruning metric at hand, when compared to the later one (b). This is because it has no extra dependency on the training data and its efficiency does not depend on the iterative re-calibration of the information state through the fine-tuning process. In this study, most experiments where conducted using "One-Shot" pruning, while we also explored the integration of NEXP in an "Iterative" pruning process with YOLOv8 (more details on 4.1), where we noted a reduction in performance declines and an improvement in the performance recovery after each pruning step, leading to better overall results.

### B.3 Detailed description of all algorithmic steps.

More in detail regarding Algorithm 1, we define the data structure $NEXP_{\text{map}}$, i.e., a dictionary in our implementation, to store the NEXP scores for every filter in the neural network after each iteration. Given a neural network $\mathcal{N}$ with its current weight state $W_{t_i}$, we initially set up all variables required for the pruning loop (Lines 1-4). The network is then gradually pruned until one of the following conditions is met: the target ratio is achieved or the allowed number of pruning steps is exceeded (Line 5). During each pruning iteration, the $\kappa$ least expressive filters from the current pruned state of the network are initially selected (Line 6). These filters are then removed, followed by an update to $NEXP_{\text{map}}$ for the subsequent iteration (Lines 7-8). To obtain the NEXP scores, a forward pass $f(X; W_{\text{pruned}})$ is conducted using a user-provided mini-batch as input. Finally, the conditions variables are updated in preparation for the next pruning iteration (Lines 9-10).

### B.4 Acceleration of NEXP computations.

In Algorithm 1, Line 8 accounts for the bulk of the computational complexity. Specifically, the calculation of $NEXP_{\text{map}}$ can be divided into two sub-processes: (i) performing a forward pass to retrieve all activation patterns, and (ii) estimating the NEXP score for each element in the map. However, performing a forward pass can be considered negligible compared to computing the NEXP score for each filter. This is because the later involves multiple comparisons between the activation patterns of all samples in the mini-batch $X$ for every filter. Two effective ways to reduce this computational demand are: first, all operations involved in computing the NEXP score are compatible with widely-used BLAS libraries, facilitating hardware acceleration; second, the frequency of score updates can be strategically decreased under certain conditions, e.g., every n pruning iterations.

## C   Experimental Settings

### C.1   Datasets and Models.

This paper explores Computer Vision tasks through extensive experiments on various datasets, such as CIFAR-10 [24] and ImageNet [40] for image classification, and COCO [33] for object detection. To demonstrate the robustness of our approach, we experiment on several popular architectures and a wide span of architectural elements, including VGGNet with a plain structure [44], ResNet with a residual structure [17], GoogLeNet with inception modules [45], MobileNet with depthwise separable convolutions [19], DenseNet with dense blocks [21] and YOLOv8 with a variety of different modules, e.g. C2f and SPPF [22].

### C.2   Adversaries.

We assess the efficacy of expressiveness as criterion for Pruning both after Training (PaT) and at Initialization (PaI), using arbitrary (random) data-points. For PaT (4.1), we compare against a plethora of foundational and state-of-the-art approaches, ranging from filter magnitude-based [28, 32, 29] and loss sensitivity-based [58] methods to feature-guided strategies [23, 30] and search algorithms [35, 31]. Regarding PaI (4.3 and D.1), our comparison is two-fold, as we evaluate expressiveness using (i) single-shot and (ii) iterative pruning. More specifically, the adversaries for PaI include pruning with random scoring, two state-of-the-art single-shot pruning strategies, namely SNIP [27] and GraSP [50], as well as one state-of-the-art iterative pruning strategy, named SynFlow [46].

### C.3   Evaluation Metrics.

To effectively quantify the efficiency of reported solutions, we adopt a 3-dimensional evaluation space, consisting of i) two widely-used metrics i.e. *FLOPs* and *params*, that define the 2-dimensional compression solution efficiency, alongside with ii) an NN model accuracy to assess the predictions of pruned derivatives [3]. Within the compression space, we define, (a) Compression Ratio($\downarrow$) $= \frac{\text{original size}}{\text{compressed size}}$ and (b) Compressed Size Percentage (%) $= \frac{\text{compressed size}}{\text{original size}} \cdot 100$. To assess task-specific capabilities, we report the top-1 accuracy of pruned models for image classification on CIFAR-10 [24], both top-1 and top-5 accuracies for ImageNet [40], and the mean Average Precision (mAP) over IoU (Intersection over Union) thresholds ranging from 0.5 to 0.95, denoted as $mAP_{50-95}^{val}$, for object detection on the COCO dataset [33].

### C.4   Configurations.

We implement the proposed "expressiveness" pruning criterion on PyTorch, version 2.0.1+cu117, by extending the DepGraph pruning framework [11] to maintain models compatibility and to ensure structural coupling during the removal of network elements e.g., simultaneously removing any inter-dependent network elements such as kernel pairs of convolutional and batch-normalization batched layers. All experiments are conducted on a NVIDIA GeForce RTX 3060 GPU with 12GB of GDDR6 SDRAM. For all experiments we use a batch of 64 random data-points to estimate expressiveness, except those that are reported for CIFAR-10 and ImageNet on 4.1, where we used K-Means to select 60 samples (6 from each class). Additionally, the baseline models on CIFAR-10 were trained for 200 epochs by using 128 batch size and Stochastic Gradient Descent algorithm (SGD) with an initial learning rate of 0.1 that is divided by 10 after 60 and 120 epochs respectively. For ImageNet models and YOLOv8, we utilize the available pre-trained weights on PyTorch vision library and ultralytics [22]. We fine-tune the pruned networks for 100 epochs on CIFAR-10 and for 30 epochs on ImageNet to compensate for the performance loss, using a batch size of 128 and 32 respectively.

## D   Supplementary Experimental Results

### D.1   Neural Expressiveness at Initialization: A comparative study

**Adversaries.** We establish our comparative study in a two-fold manner, as we compare expressiveness against (i) single-shot and (ii) iterative pruning approaches. More specifically, the adversaries include pruning with random scoring, two state-of-the-art single-shot pruning strategies, namely SNIP [27]

and GraSP [50], as well as one state-of-the-art iterative pruning strategy, named SynFlow [46]. For our approach, we implement one-shot pruning, utilizing a batch of 64 arbitrary data points for the estimation of expressiveness.

**Experimental Setup.** We adopt the experimental framework of Tanaka et al. [46], who assess algorithm performance across an exponential scale ($10^r$) of parameters compression ratios $r \in \{0.00, 0.25, 0.50, 0.75, \dots\}$. Their proposed settings also enable for the evaluation of an algorithm's resilience to "layer collapses", typically observed at higher compression levels. **Results.** We prune VGG-16 on CIFAR-10 and compare against the findings of [46]. We remain consistent with our adversaries and train the model for 160 epochs, using a batch size of 128 and an initial learning rate of 0.1, which is reduced by a factor of 10 after 60 and 120 epochs. The results are illustrated on Fig. 5.

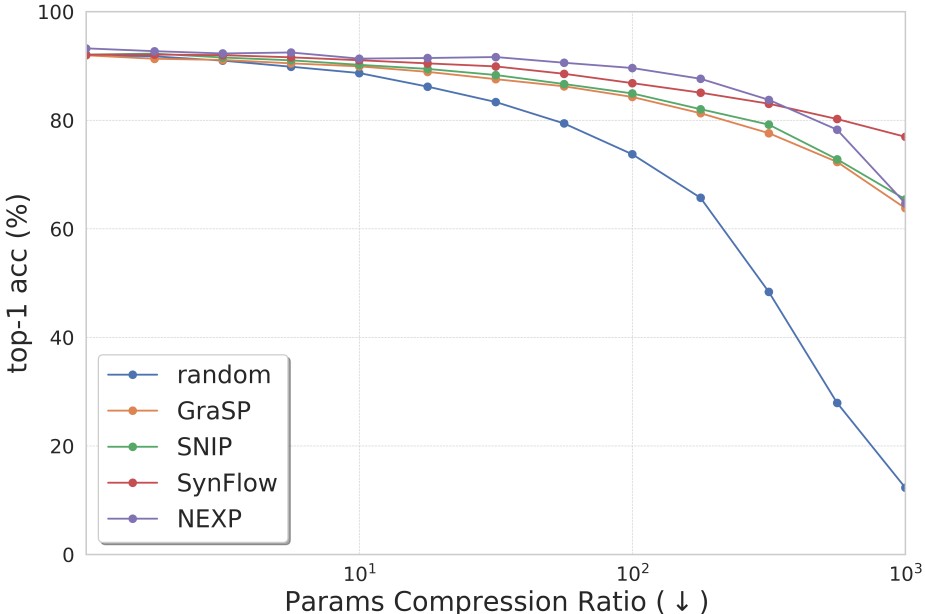

Figure 5: **Pruning VGG-16 at Initialization on CIFAR-10.** A comparative visualisation of SOTA methods across an exponential scale of params compression ratios.

**Observations.** Our method consistently outperforms all other algorithms, particularly in regimes of lower compression, up to $10^2(\downarrow)$ with an average increase of 1.21% over SynFlow, while maintaining competitiveness at higher compression levels, above $10^2(\downarrow)$ with an average percentage difference of 4.82%, 3.72% and -2.74%, compared to GraSP, SNIP and SynFlow respectively.

## D.2 Additional Experimental Results: Tables and Figures

**CIFAR-10.** We present further experiments and comparisons with state-of-the-art methods, including HRANK [30], GAL [32], ABC [31] and DCP [61], specifically for GoogLeNet and MobileNet-v2 networks. For MobileNet-v2, our method attains an increased compression ratio of $0.94\times$ in parameters and $0.75\times$ in FLOPs ($\downarrow$), with a minimal decrease of only -0.09% in performance compared to DCP. In the GoogLeNet case, we demonstrate a notable enhancement in parameters compression within the $1.60\times$ to $2.20\times$ FLOPs compression range, surpassing GAL and HRANK with margins of $1.8\times$ and $1.52\times$ respectively, with an average improvement of 7.5% in performance degradation.

Table 6: Analytical Comparison of Importance-based solutions and Expressiveness on CIFAR-10 using VGGNet architectures [44].

| Model | Method | top-1 acc | | Compression Ratio ↓ | |
| | | Base (%) | Δ (%) | #Params | #FLOPs |
|---|---|---|---|---|---|
| VGG-16 | L1 [28] | 93.25 | +0.15 | 2.78× | 1.52× |
| | GAL-0.05 [32] | 93.96 | -0.19 | 4.46× | 1.65× |
| | GAL-0.1 [32] | | -0.54 | 5.61× | 1.82× |
| | HRank [30] | 93.96 | -0.53 | 5.97× | 2.15× |
| | HRank [30] | 93.96 | -1.62 | 5.67× | 2.89× |
| | SCP [23] | 93.85 | -0.06 | 15.38× | 2.96× |
| | NEXP (Ours) | 93.87 | -0.16 | 5.62× | 3.03× |
| | ABC [31] | 93.02 | +0.06 | 8.80× | 3.80× |
| | NEXP (Ours) | 93.87 | -0.35 | **13.13×** | 4.01× |
| | HRank [30] | 93.96 | -2.73 | 8.41× | 4.26× |
| VGG-19 | DCP-Adapt [61] | 93.99 | +0.58 | 15.58× | 2.86× |
| | SCP [23] | 93.84 | -0.02 | 20.88× | 3.86× |
| | NEXP (Ours) | 94.00 | -0.53 | **22.73×** | 4.75× |

Table 7: Analytical Comparison of Importance-based solutions and Expressiveness on CIFAR-10 using GoogLeNet [45].

| Model | Method | top-1 acc | | Compression Ratio ↓ | |
| | | Base (%) | Δ (%) | #Params | #FLOPs |
|---|---|---|---|---|---|
| GoogLeNet | GAL-0.5 [32] | 95.05 | -0.49 | 1.97× | 1.62× |
| | NEXP (Ours) | 94.97 | -0.43 | **3.77×** | 2.12× |
| | Hrank [30] | 95.05 | -0.52 | 2.25× | 2.20× |
| | ABC [31] | 95.05 | -0.21 | 2.51× | 2.99× |
| | NEXP (Ours) | 94.97 | -1.07 | **7.02×** | 3.01× |
| | Hrank [30] | 95.05 | -0.98 | 3.31× | 3.38× |

Table 8: Analytical Comparison of Importance-based solutions and Expressiveness on CIFAR-10 using DenseNet-40 [21].

| Model | Method | top-1 acc | | Compression Ratio ↓ | |
| | | Base (%) | Δ (%) | #Params | #FLOPs |
|---|---|---|---|---|---|
| DenseNet-40 | GAL-0.5 [32] | 95.05 | -0.49 | 1.97× | 1.62× |
| | Hrank [30] | 95.05 | -0.52 | 2.25× | 2.20× |
| | NEXP (Ours) | 94.64 | -0.89 | **2.72×** | 2.25× |
| | NEXP (Ours) | 94.64 | -0.84 | **3.12×** | 2.51× |
| | ABC [31] | 95.05 | -0.21 | 2.51× | 2.99× |
| | Hrank [30] | 95.05 | -0.98 | 3.31× | 3.38× |

Table 9: Performance Outcomes for MobileNet-v2 on the CIFAR-10 Dataset.

| Method | Base (%) | Δ Acc (%) | #Params ↓ | #FLOPs ↓ |
|---|---|---|---|---|
| DCP [61] | 94.47 | +0.22 | 1.31× | 1.36× |
| NEXP (Ours) | 94.32 | +0.13 | 2.25× | 2.11× |

**YOLOv8.** Figure 6 compares Neural Expressiveness (NEXP) with Layer-Adaptive Magnitude-Based Pruning (LAMP) [26], Network Slimming (SLIM) [35], Wang et al.'s DepGraph [11], and Random Pruning for Object Detection on the COCO dataset, as discussed in 4.1.

**Motivation.** YOLOv8 [22] is the current state-of-the-art for Object Detection and Image Segmentation, and has already been widely adopted by many for a variety of real-time applications, e.g. Traffic Safety [1], Medical Imaging [39], Rip Currents Detection [10], and more. Such applications could majorly benefit from model compression optimizations, achieving higher throughput ratios that translate to increased resolution (FPS), and enabling deployment on hardware with strict resource constraints.

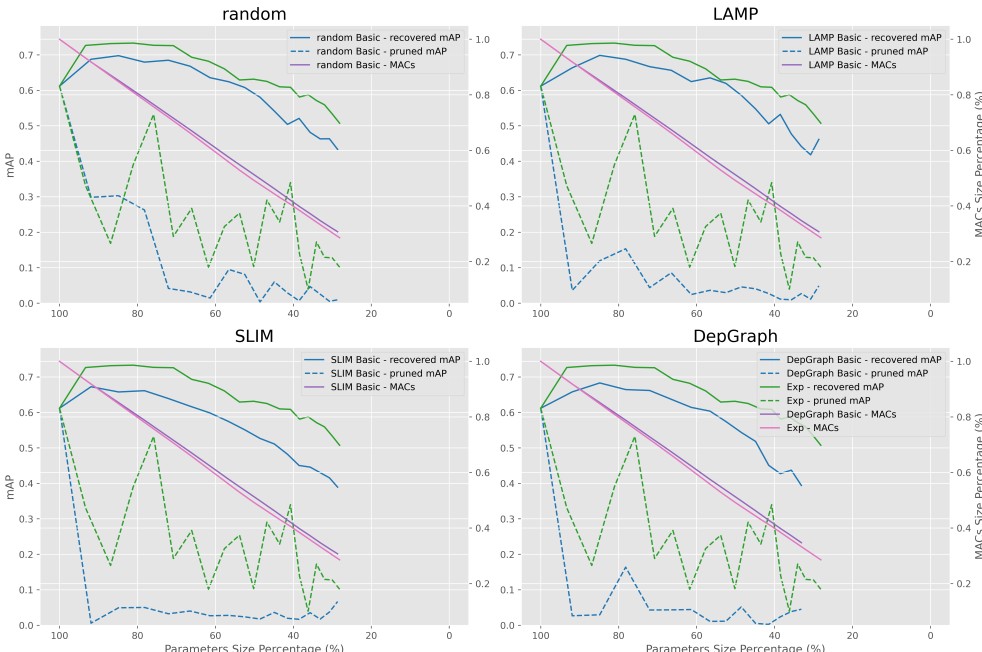

Figure 6: **Pruning YOLOv8m trained on COCO for Object Detection.**

