# OpenReview forum: "Neural expressiveness for beyond importance model compression"
_NeurIPS.cc/2024/Conference — Submitted to NeurIPS 2024_

### Official Review · Reviewer_pBA2 · 2024-07-07

**Soundness:** 2
**Presentation:** 2
**Contribution:** 2
**Rating:** 5
**Confidence:** 4

**Summary:**

In this paper, the author proposes "Expressiveness," a metric that measures the dissimilarity of feature maps produced by different filters. Subsequently, the author introduces NEXP, a technique to prune filters based on their expressiveness. The proposed method is tested on tasks such as image classification and object detection.

**Strengths:**

1. The overall structure of the paper is clean and easy to follow.
2. Although I am not very familiar with the structured pruning literature, the application of the concept of representation power of neural networks in this field appears novel.
3. The experiments are comprehensive, including pruning at initialization (PaI), pruning after training, and other tasks like object detection.

**Weaknesses:**

1. The notation is very confusing. For example, in the section 'Generalization of concepts at a structural level,' $\ell$ is the index of a certain layer, but the upper-case K is the total number of layers, and the lower-case k is the index of a filter/channel in a certain layer.
2. The concept of expressiveness is not new in the context of pruning [1] and neural architecture search [2,3,4].
3. The performance improvement by NEXP is not prominent. For example, in Tables 1 and 2, NEXP often shows a higher compression ratio but lower accuracy. This makes it unclear if NEXP offers a significant advantage over other methods.

***
**Minor Mistakes:**

Line 163: "where k the is" should be corrected.

[1] Tanaka, Hidenori, et al. "Pruning neural networks without any data by iteratively conserving synaptic flow." Advances in Neural Information Processing Systems 33 (2020): 6377-6389.

[2] Lin, Ming, et al. "Zen-NAS: A zero-shot NAS for high-performance image recognition." Proceedings of the IEEE/CVF International Conference on Computer Vision. 2021.

[3] Wang, Haoxiang, et al. "Global convergence of MAML and theory-inspired neural architecture search for few-shot learning." Proceedings of the IEEE/CVF Conference on Computer Vision and Pattern Recognition. 2022.

[4] Chen, Wuyang, Xinyu Gong, and Zhangyang Wang. "Neural architecture search on ImageNet in four GPU hours: A theoretically inspired perspective." arXiv preprint arXiv:2102.11535 (2021).

**Questions:**

In section D.1, how is NEXP applied to PaI methods? If I understand correctly, SNIP, GraSP, and SynFlow are all weight pruning methods, whereas NEXP is structured/filter pruning method.

**Limitations:**

The authors have discussed limitation in the manuscript.

---

> ### Author Rebuttal · Authors · 2024-08-05
>
> **W1.** We have adjusted the notation formatting in 3 and especially in "Generalization of concepts at a structural level" to improve clarity. Below is an analytical list of all changes made in the paper:
>
> W1.a. Changes emphasized on the "Generalization of concepts at a structural level":
> - Lines 160-161: The notation $K$ has been replaced by an analytical expression for the convolutional layers.
>
>     **Updated Sentence:** For a CNN model with a set $\textcolor{blue}{C = \\{C^{1}, \dots, C^{l}, \dots, C^{|C|}\\}}$ of $\textcolor{blue}{|C|}$ convolutional layers, where $C^{l}$ is the $l$-th convolutional layer.
>
> - Lines 162-163: Based on the fact that the volume of the activations output equals the number of filters applied to a given input because of the intrinsic properties of the convolutional operations (i.e., $ |F^{l}| = |C^{l}| $), we extend the sentence in the following way:
>
>     **Updated Sentence:** We denote filters (weight maps) and  feature maps (activation maps) as $F_{k}^{l}$ and $C_{k}^{l}$, respectively, where $k$ $\textcolor{blue}{\text{is the index within a layer}}$ and $\textcolor{blue}{|F^{l}| = |C^{l}|}$. *(Thank you for your attention to detail regarding minor mistake 1.)*
>
> - Eqs 4 and 5: $K^{(1)}$ and $K^{(l)}$, which represent the number of filters in layer 1 and a given layer $l$, respectively, have been replaced by the cardinality notation, i.e., $|C^{(1)}|$ and $|C^{(l)}|$.
>
> W1.b. To maintain a more coherent notation structure throughout the entire section, we apply the following changes to the remaining of Section 3:
> - Line 172-175: The sentence has been polished to incorporate the updated notation for convolutional layers as introduced above. Specifically,
>
>     Line 173: *"a CNN model with $K$ convolutional layers"* **has been updated to** *"a CNN model with $\textcolor{blue}{C}$ convolutional layers"*.
>
>     Line 174: $F^{|K|}$ **has been updated to** $F^{\textcolor{blue}{|C|}}$ and $F^{(l, K^{(l)})}$ **has been updated to** $F^{(l,\textcolor{blue}{|C^{(l)}|})}_{t_i}$. (Note: Subscripts are not displayed correctly due to the OpenReview LaTeX formatting environment.)
>
>     Line 175: *"with $K^{(l)}$ being the amount of weight maps (filters) in a given layer $l$."* **has been updated to** *"with $\textcolor{blue}{|C^{(l)}|}$ being the $\textcolor{blue}{number}$ of weight maps (filters) in a given layer $l$."*
>
> -  In the remainder of Section 3, we adopt the more consistent and clearer notation structure using $C$, as refined in W1.a and W1.b. Specifically, we replace the $K$ notations with the more appropriate $C$-based notations in lines 184, 189-190, 210-211, and in eqs 6 and 7.
>
> W1.c. see reviewer 4Yxj W2 rebuttal response.
>
> **W2.** Indeed, the concept of expressiveness, or else discriminative capacity of neuron activations, has been discussed in other papers, especially in the domain of Neural Architecture Search, as a more prominent solution for evaluating the capacity of neural networks. However, this is the first work to establish a concrete categorization between incorporating weights and activations in model compression decisions. Specifically, to the best of our knowledge, this work is the first to provide:
> - (a) a distinct motivation for exploring activation-based compression approaches (1), as the current state of the literature is predominantly focused on weight-based methods, with recent research interest increasingly shifting towards activations [1].
> - (b) a detailed categorization and examination of the motivations and limitations of current literature on pruning weight and discriminative -based methods (2).
> - (c) an in-depth mathematical conceptualization of neuron activations properties and the benefits of **"information flow"** for model compression in a model-agnostic format that can be incorporated into any activation-based approach (3), while it demonstrates the complementary benefits of weight-based (importance) and activation-based (expressiveness) pruning methods, highlighting their partial orthogonality (4.2).
>
> Overall, the proposed concept of **"Neural Expressiveness"** in this paper expands on the prevalent yet unexplored notion of expressiveness in efficient deep learning and aims to forge a distinct paradigm, transitioning from the conventional weight-centered importance assessment to an emphasis on activations.
>
> **W3.** Overall, this work suggests that no universal solution outperforms all compression methods across all setups and contexts, as subjective quality metrics (e.g., accuracy) are often difficult to replicate and may vary significantly for different setups (see lines 274-277) [2]. Specifically, the efficiency of model compression methods can be directly assessed by metrics like FLOPs and parameters, while predictive quality is influenced by factors such as post-pruning fine-tuning configurations, pruning process design parameters (B.2), etc... [2]. For that reason, the experimental section emphasizes the proposed method's applicability across various settings: (a) "one-shot" or "iterative" pruning (4.1 and B.2), (b) PaT and PaI (4.1, 4.3, and D.1), and (c) hybrid solutions (4.2). This provides a foundation for further optimization in specific tasks. Overall, NEXP using iterative pruning shows notable performance improvements for object detection (lines 287-315) and one-shot pruning for PaI (4.3), while consistently improving compression across all tables and maintaining performance, with notable improvements in some regimes, e.g., Table 1 (right) +0.66, +0.10.
>
> Also kindly refer to Q2 in 9m28.
>
> **Q1.** It is applied similarly to PaT (4.1), with the only difference being that the network $\mathcal{N}$ in Alg.1 is untrained (see Section A). The network is pruned in a one-shot manner (672-674) using NEXP for given compression ratios $r = \tau$ (672-677) and then trained as outlined in lines 679-681.
>
> [1]. Liu, Ziming, et al. "Kan: Kolmogorov-arnold networks." arXiv preprint arXiv:2404.19756 (2024).
>
> [2]. ref [3] in paper

---

> > ### Comment · Reviewer_pBA2 · 2024-08-11
> > **Thank you for your rebuttal.**
> >
> > Thank you for your detailed rebuttal. After reviewing the authors' response and reviews from other reviewers, my primary concern W3 remains unchanged. The experimental evaluation presented in the manuscript demonstrates that while NEXP can be applied to many scenarios, it is often not the best-performing method when compared to the baselines. Or in some cases, the comparisons do not show which method is superior.
> >
> > Put simply, NEXP may be a good approach, however, the evidence provided does not convincingly show that it offers a significant improvement over existing baselines. Therefore, I do not find sufficient justification to revise my initial rating. As a result, I will be maintaining my original rating.
> >
> > Thank you again for your efforts.

---

> > > ### Author Response · Authors · 2024-08-13
> > > **Addressing Concerns Regarding Performance Improvements (part a)**
> > >
> > > Thank you for taking the time and effort to address our initial rebuttal (especially at this time of year). Your insightful comments and suggestions are greatly appreciated and have enabled us to further clarify the motivations of our work and improve the presentation quality of the paper.
> > >
> > > To further address your concerns regarding performance improvement by NEXP (W3), we would like to reiterate some of the key points from the paper, with emphasis on the experimental section, for clarity.
> > >
> > > The experimental setup and structure of this work are motivated by our understanding that the pruning criteria for each compression use case scenario may vary significantly, e.g., availability of computational resources, cost of deployment, specifications of the target hardware device, availability of data, target quality and efficiency of the pruned model, and many more. Below, we present some specific (1 and 2) and general (3, 4, and 5) examples to highlight the intricacies of defining the (combinatorial) pruning constraints during the compression process and to demonstrate the effectiveness of NEXP beyond the predictive performance (quality) of the pruned models.
> > >
> > > 1. DCP employs 400 training epochs for fine-tuning post-pruning to recover predictive performance, compared to our method, which uses only 100 epochs (as outlined in lines 276-277). This may result in a significant increase in operational costs for cloud-based hardware infrastructures. Specifically, given that the pruned models generated by both NEXP and DCP have similar compression sizes and inference speeds, DCP will require $4\\times$ more funds to generate the pruned model compared to NEXP.
> > >
> > > 2. HRANK utilises 500 input samples to estimate the average rank, while Network trimming [3] requires over ten thousand input
> > > images to estimate the sparsity of feature maps, compared to ours random selection of 64 for the estimation of NEXP. This results to a $\\approx8\\times$ reduction in dependency on input data compared to HRANK and several orders of magnitude less than [3].
> > >
> > > 3. NEXP requires only forward passes, unlike gradient-based pruning methods that also require backward passes, which are also intrinsically linked to the availability and quality of data. In contrast, NEXP is independent of both the quality and quantity of samples, as demonstrated in Appendix A and discussed in Section 3.3 (*"Dependency to Input Data"*) of the paper.
> > >
> > > 4. NEXP maintains consistent performance without a significant drop in computational efficiency as models and tasks grow in complexity. In contrast, weight-based pruning methods incur increased computational overhead as model sizes increase, as discussed in W2 of reviewer 9m28, primarily due to their dependence on the dimensions and cardinality of layer weight matrices. Unlike these methods, NEXP's efficiency is not correlated with model size, demonstrating strong scalability as model complexity increases.
> > >
> > > 5. NEXP yields consistent estimations across the evolutionary (learning) stages of a neural network, making it a suitable criterion for pruning both untrained and trained networks, as demonstrated in Subsection 4.3 and discussed in Appendix A. To the best of our knowledge, our proposed approach is the first criterion designed for both Pruning at Initialization (PaI) and Pruning after Training (PaT). This stands in contrast to other fundamental principles of pruning metrics in the literature, which are specifically designed to address only one of these challenges and do not extend across the full spectrum of the convergence process.
> > >
> > > Next, we also present 3 specific examples from the experimental section to highlight the performance improvement achieved by NEXP:
> > >
> > > 1. Object Detection with YOLOv8: Our work demonstrates the superiority of NEXP for the more complex task of object detection, yielding significant performance improvements across the entire pruning spectrum compared to all adversary methods (as shown in Figs 2 and 6). Specifically, in lines 302-315, we discuss the superiority of our method using an iterative pruning format to further highlight the intrinsic property of expressiveness in maintaining network elements that are more robust to information redistribution during re-training, in contrast to the 'important' labeled structures identified by other methods. The experimental sections are arranged in this order to address the potential limitations of employing one-shot pruning for NEXP, as demonstrated in the image classification experiments in the previous subsection.
> > >
> > > 2. NEXP at Initialization: NEXP consistently outperforms all other approaches in top-1 accuracy, particularly in parameters compression regimes with up to $100\\times$ smaller networks.
> > >
> > > 3. Tables 1 and 2 for image classification: we kindly ask you to refer to part b of this response.
> > >
> > > [3]. Hu, Hengyuan, et al. "Network trimming: A data-driven neuron pruning approach towards efficient deep architectures." arXiv preprint arXiv:1607.03250 (2016).

---

> ### Author Response · Authors · 2024-08-13
> **Addressing Concerns Regarding Performance Improvements (part b)**
>
> 3. Thank you for acknowledging NEXP's superiority in consistently achieving higher compression ratios. While it is true that in some cases, as seen in Tables 1 and 2, the predictive quality of models generated by NEXP is not always superior to other methods, it is equally important to note the numerous instances where NEXP demonstrates notable performance improvements (e.g., Table 1, right: lines 1-3, 8-9; Table 1, left: lines 10-11; Table 2: lines 6-8). In this context, we believe that a fair comparison, which includes cases where our approach does not outperform alternative methods, offers a valuable basis for a more thorough discussion and for identifying ways to address potential limitations. In this context, as also discussed in point 1, we address the potential limitations in predictive performance by providing a detailed discussion on the advantages of using an iterative pruning format (as demonstrated in object detection, where NEXP shows significant performance improvements) compared to the one-shot pruning format employed for image classification in Tables 1 and 2. A more in-depth analysis of the pruning settings is also available in Appendix B ('Pruning Process: An In-Depth Analysis').
>
> Overall, we concur with the notion that the pruned architecture, rather than the inherited 'important' weights, is more crucial to the efficiency of the final model, as discussed in Q2 of reviewer 9m28 and outlined in [4]. However, under different experimental settings and evaluation criteria (e.g., Tables 1 and 2), 'important' weights may prove to be a superior metric for achieving higher predictive performance, especially when predictive accuracy is a more critical constraint than parameter reduction. To address this potential trade-off between important and expressive methods along the efficiency-quality curve, we have included an assessment of the hybrid compression space (Subsection 4.2). This assessment highlights the partial orthogonality of the two compression criteria (i.e., importance of weights and expressiveness of neurons) and provides both an intuitive understanding and a solid algorithmic basis for hybrid optimizations, and thus enables a more comprehensive exploration of the trade-off between the quality of the pruned architecture and the inherited 'important' weights (*in our opinion, an open and intriguing challenge in the domain of model compression*). To the best of our knowledge, this is the first work to propose hybrid solutions for model compression.
>
> We are more than willing to address any further questions or concerns you might have to ensure that our submission meets the highest quality standards.
>
> [4]. Liu, Zhuang, et al. "Rethinking the value of network pruning." arXiv preprint arXiv:1810.05270 (2018).

---

### Official Review · Reviewer_ha2w · 2024-07-09

**Soundness:** 2
**Presentation:** 2
**Contribution:** 2
**Rating:** 4
**Confidence:** 4

**Summary:**

This paper works on weight pruning for CNNs. It proposes an evaluation metric, i.e., "expressiveness", to evaluate whether a neuron/groups of neurons should be pruned or not. The metric focuses on the neurons' ability to redistribute informational resources. As the evaluation of expressiveness requires data samples, the paper includes studies on arbitrary data or limited dataset’s representative samples. The experiments are conducted for image classification tasks on ResNet architectures, and the object detection task on YOLOv8m.

**Strengths:**

Weight pruning is an effective manner in reducing the redundancies in DNNs. Instead of focusing on weight importance, this paper considers the expressiveness of neurons in the information flow within a network. The proposed evaluation metric can also be combined with existing strategies with importance evaluation metrics as a hybrid pruning approach.

**Weaknesses:**

1. Limited practicality of the approach. The method is mainly focused on removing redundant filters from CNNs. Though CNN is one category of DNNs, recent works have shifted to more advanced model architectures such as transformers and Mamba, which are mainly composed of FC layers instead of CNNs. The practicality of the approach is highly restricted to SOTA model architectures for image classification tasks.

2. Limited performance improvements. This is a major concern. The performance gain of the proposed method is not obvious compared with baselines. For instance. on CIFAR-10 VGG-16, SCP and reduce the parameters 15.28$\times$ with a 93.85\% accuracy while the proposed method can only reduce the parameters 5.62$\times$ with a slightly better accuracy 93.87\%. HRank also provides better performance than the proposed method with higher accuracy 93.96\% (0.09\% higher than NEXP) and higher reductions of FLOPs (4.26$\times$ (HRank) v.s. 4.01$\times$ (NEXP) ). On DenseNet-40, Hrank also shows better performance across all metrics. Hrank v.s. NEXP: Accuracy 95.05\% v.s. 94.64\%, parameter reduction 3.31$\times$ v.s.3.12$\times$, FLOPs reduction 3.38$\times$ v.s. 2.51$\times$.

**Questions:**

1. Can this method be extended to model architectures beyond CNNs? How is the performance compared with other methods?

2. What is the acceleration performance of the proposed method?

3. For the data selection, in extreme cases, such as if the data samples all come from the same classification class, what will the performance be like?

**Limitations:**

Please refer to weaknesses and questions. The major concern is that the method does not show better performance than baselines, and is also limited in model architectures.

---

> ### Author Rebuttal · Authors · 2024-08-06
>
> **W1 and Q1.** NEXP is designed to measure the redundancy of activation structures based on their expressiveness, where the finest granularity can be considered that of a single neuron, and thus it can be adjusted to any activation structural component, e.g., convolutional filters (as demonstrated in the paper). In Section 3, we selected the convolutional operation to illustrate the intrinsic properties of NEXP for structured activation patterns, recognizing it as the most widely used and fundamental operation in recent years (lines 159-161 and 193). However, our approach is designed to facilitate structural pruning across all neural networks. This is achieved by extending the intuitions and fundamentals presented in Section 3 and adjusting Eq.11 accordingly by substituting the filter ($F_{k}^{l}$) with the neural network element of choice. More specifically, some of the architectures that NEXP is currently compatible with for pruning by extending the latest version of DepGraph [1] include Large Language Models (LLMs), Segment Anything Model (SAM), Diffusion Models, Vision Transformers, ConvNext, Yolov7, Yolov8, Swin Transformers, BERT, FasterRCNN, SSD, ResNe(X)t, DenseNet, RegNet, and DeepLab.
>
> **W1.** The paper includes an evaluation and analytical discussion of the proposed approach (NEXP) on the SOTA YOLOv8 model architecture for object detection (lines 287-315 in Section 4.1), where NEXP demonstrates superior performance in terms of both compression efficiency and predictive quality for the pruned models. Additionally, NEXP has been employed to prune the FC layers in vision architectures. For instance, ResNet-56 contains an FC layer before the classification layer, which was pruned as part of the evaluation process reported in Table 1. In more detail, Table 1 in the 1-page rebuttal PDF provides an analytical comparison between the baseline and pruned ResNet structures using NEXP, focusing on the last block and the FC layer (for the pruned model of row 8 in Table 1 of the paper).
>
> **W2.** In Section 4.1, the results of adversarial pruning methods for image classification are sourced directly from the respective publications (please refer to the response in W3 for our rebuttal to reviewer's comments 9m28). To ensure a fair and consistent comparison of predictive quality across all various pruning approaches (as shown in Tables 1, 2, and 6-9), this work presents both the predictive performance of the unpruned (baseline) models, labeled as Base (%), and the pruned models, labeled as ∆ (%), as reported by each study.  Therefore, the acc-1 percentages in W2's comparisons mistakenly refer to the accuracy of the unpruned models rather than the accuracy of the pruned models produced by each method. To improve the clarity of the paper, we have extended lines 270-272 to include a description of the predictive quality notation as follows: *“The results of adversarial pruning methods in this subsection are directly obtained from the respective publications. In Tables 1, 2, and 6-9, the predictive performance of the unpruned (baseline) models, labeled as Base (%), and the pruned models, labeled as ∆ (%), are presented, along with their respective compression ratios, as reported by each study."*.
>
> Addittionally, kindly refer to the response in W3 for the rebuttal to reviewer's pBA2 comments and to Q2 for the rebuttal to reviewer's 9m28.
>
> **Q2.** The term *'acceleration performance'* of the proposed method can be ambiguous, as it may refer to both the computational efficiency of the pruning process and the accelerated performance of the pruned networks. To address both aspects, we provide clarifications below. Please feel free to further clarify your statement if we have misinterpreted it in the comments section.
>
> Q2.a. **Acceleration of NEXP computations:** Kindly refer to the response in W2 for the rebuttal to reviewer's 9m28 comments.
>
> Q2.b. **Acceleration of pruned networks:** To effectively quantify the efficiency of reported solutions, this work emphasizes on theoretical compression metrics, i.e. (a) the number of multiply-adds (referred to as FLOPs) required to perform inference with the pruned network and (b) the fraction of parameters pruned, and does not include evaluations of the practical runtime speedup of pruned models. We acknowledge the significance of such evaluations and the limitations of theoretical proxies [2]. While all parameters (which are emphasized in this work) can be treated equally when reducing the network's storage footprint, different parameters may have varying impacts when reducing the computational cost of inference. However, this work is structured to provide a consistent comparative framework based on more accessible and common theoretical metrics, ensuring ease of comparison with other approaches. For that reason, following the intuition of Blalock, Davis, et al. [2], we provide a comprehensive detailing of the efficiency metrics in C.2, with further analytical discussions provided throughout the paper, e.g. lines 23-36, 3.4, Alg.1, and section 4 overall.
>
> **Q3.**  Our initial intuition was that discriminative and activation -based approaches rely on sample quality. However, the sensitivity analysis of NEXP in A and 3.3 highlight that NEXP is better approximated using random samples, rather than class-representative sampling via k-means. Although extreme cases were not evaluated, most experiments used small batches of random samples to estimate Neural Expressiveness (lines 659-661). In this context, the evaluation on ImageNet-1k (4.1), using 64 random samples across 1000 classes, is a prominent indicator that NEXP efficiency remains consistent in such scenarios.
>
> [1]. Fang, Gongfan, et al. "Depgraph: Towards any structural pruning." Proceedings of the IEEE/CVF conference on computer vision and pattern recognition. 2023.
>
> [2]. Blalock, Davis, et al. "What is the state of neural network pruning?." Proceedings of machine learning and systems 2 (2020): 129-146.

---

> > ### Comment · Reviewer_ha2w · 2024-08-13
> >
> > I thank the authors' for providing the response to my questions. After reading the rebuttal, I still have concerns towards: 1) The application scenarios towards other model architectures, especially to more latest building blocks such as transformer blocks with more fully connected layers rather than convolutional layers. There is no evidence how the method will perform to these models. 2) The performance improvements. Though the reported accuracy is the base accuracy of the dense model, the performance improvements comparing to baselines are not obvious. For instance, in table 1, ABC only degrades the accuracy by 0.03\% with 2.18$\times$ parameters reduction and 2.18$\times$ FLOPs reduction. However, NEXP degrades the accuracy by 0.41\%, with 2.11$\times$ FLOPs reduction and 2.87$\times$ parameters reduction.
> >
> > Thus, I would keep my current rating and thanks the authors' efforts.

---

> ### Author Response · Authors · 2024-08-13
> **Addressing Reviewer ha2w's Concerns Regarding W1 and W2**
>
> We would first like to thank you for taking the time and effort to address our initial rebuttal (especially at this time of year).  Your insightful comments and questions are greatly appreciated and have enabled us to further clarify the motivations of our work and improve the presentation quality of the paper.
>
> To further address your concerns regarding the applicability (W1) and performance improvements (W2) of our proposed method, we provide more detailed discussions below.
>
> **W1 (extended).** Indeed, the paper does not provide experimental evidence on how our proposed approach will perform in transformer blocks. While we have incorporated our proposed pruning approach as an extension of the latest DepGraph version (and thus its pruning compatibility aligns with that of the DepGraph framework) and have provided theoretical 'guarantees' via the generalization of the foundational concepts and intuitions of Neural Expressiveness—i.e., estimating the contribution of a neuron or group of neurons based on their ability to extract features that maximally separate sub-spaces within the feature space using the overlap of activations—we believe that, in order to perform a robust analysis and provide concrete experimental evidence, the intricacies of the various operations handling computational workloads within different neural network architectures should be explored and discussed separately. For that reason, the format of this paper prioritizes providing a solid basis on both the theoretical and algorithmic properties of NEXP, while hinting at potential future directions. To motivate further research and analysis of NEXP in task-specific and model-specific optimizations, we kindly encourage readers to experiment with different settings of NEXP throughout the paper, and we provide specific guidelines (e.g., lines 219-222, 234-236).
>
> **W2 (extended).** For an in-depth discussion on the performance improvements achieved by NEXP, as well as its effectiveness beyond the predictive performance (quality) of the pruned models, we kindly refer you to the responses titled *'Addressing Concerns Regarding Performance Improvements (part a)'* and *'Addressing Concerns Regarding Performance Improvements (part b)'* in reviewer pBA2's comments section.
>
> W2.a. Addressing the Example on ABC: The ABCPruner can be categorized as part of the subset of Evolutionary-Based Search Algorithms within the domain of Model Compression [3]. In contrast to most pruning methods, which focus on removing redundant network elements—whether weights, neurons, or structures of weights and/or neurons— Neural Architecture Search (NAS) involves searching through a vast space of possible architectures to find the optimal one, often requiring the training of many candidate models from scratch. Specifically, NAS-Evolutionary-Based pruning approaches adopt evolutionary algorithms to explore and identify optimal sparse subnetworks; in this context, ABCPruner employs the artificial bee colony algorithm to efficiently discover the optimal pruned structure. While NAS-based algorithms often do not constitute a fair comparison to pruning methods due to their more intensive and costly explorations (when compared to the more lightweight and straightforward intuition of elimination in pruning approaches), we have included ABC as a more generic compression baseline to further emphasize NEXP's consistent ability to achieve higher compression ratios beyond those of native pruning methods.
>
> We are more than willing to address any further questions or concerns you might have to ensure that our submission meets the highest quality standards.
>
> [3]. He, Yang, and Lingao Xiao. "Structured pruning for deep convolutional neural networks: A survey." IEEE transactions on pattern analysis and machine intelligence (2023).

---

### Official Review · Reviewer_9m28 · 2024-07-13

**Soundness:** 2
**Presentation:** 3
**Contribution:** 2
**Rating:** 5
**Confidence:** 4

**Summary:**

This paper propose a new structured pruning approach NEXP. It works by computing the dissimilarity score of the feature activations across samples and removing those filters with smaller variances. Experimental results on several models and datasets demonstrate the effectiveness of the proposed approaches.

**Strengths:**

1. The paper is well-written with clear motivation and many of the important technical details are included.
2. The authors have presented the experimental results well and the additional discussion provides deeper insights on the effectiveness of the proposed methods.
3. The authors evaluated models beyond image classification, i.e., the proposed methods work well on YOLOv8 object detectors.

**Weaknesses:**

1. The conclusion section is too short and fails to characterize the main contribution of this paper. What does it mean as to “when” and “how” to prune? The authors should elaborate on these further.

2. I think that this is not a scalable approach for computing the pruning metrics. The pruning metric proposed in the paper requires computing a N by N matrix for each filter in the network, where N is the number of samples. This could grow quite computationally infeasible for large networks and batch sizes. The authors also fails to discuss this aspect on the pruning efficiency in the paper.

3. In terms of experiments, I am not sure why the authors compare each methods under different compression ratio? If the pruned models in each method have different parameters, it can be hard to compare the accuracy numbers.

4. I feel a lot of the content in section 3 is not necessary and they can could go into a separate preliminary section. Section 3.1 and early parts of Section 3.2 takes up a lot of space and in the meantime do not provides us with the motivation and insights for the later introduced methods.

**Questions:**

1. The authors mostly focus on convolutional neural networks. However, Vision Transformers are becoming the de facto architecture for many computer vision tasks. Does the proposed approach apply to Vision Transformers as well?

2. Figure 3 seems to suggest that the proposed pruning metric is uniformly superior to importance based weight metric. Why is this the case? It seems unintuitive to me that a pruning metric that discard the weight information could works better than magnitude based approaches.

3. Have the authors evaluate the practical runtime speedup of pruned models? This would make the results more comprehensive.

4. What does Figure 1 depicts exactly? What does each figure represent, e.g., the row and column?

**Limitations:**

Yes.

---

> ### Author Rebuttal · Authors · 2024-08-06
>
> **W1.** Kindly refer to the response in W1 for the rebuttal to reviewer's 4Yxj comments.
>
> **W2.** The proposed pruning metric, as demonstrated in 3.2 and illustrated in eq.11, requires a total of ${\frac{N(N-1)}{2}}$ combinations, unlike the $N^{2}$ combinations in an $N$ by $N$ matrix, as highlighted and discussed in lines 225-233 of Subsection 3.2. We acknowledge and agree that computational efficiency is a crucial aspect of pruning approaches. Accordingly, our submitted paper includes a dedicated subsection (B.4) on the **'Acceleration of NEXP computations**', as referenced in lines 264-265. To improve the clarity of App. B's contents in the main text, we have refined the sentence in lines 264-265 as follows:
>
> A more in-depth analysis of Alg.1 along with more details on the implementation options, $\textcolor{blue}{\text{i.e., B.1 Global vs local -scope pruning, B.2 One-shot vs Iterative pruning, B.3 Detailed description of all algorithmic steps, and B.4 Acceleration of NEXP computations,}}$ are presented in Appendix B.
>
> Following, we also provide a comparison between the average results of 100 pruning iterations using $l1$ magnitude pruning  (which is consider a lightweight approach) and NEXP as implemented in the DepGraph pruning framework for CIFAR-10 and ImageNet, following the experimental protocol outlined in the paper. We present the average duration of a pruning step along with its standard deviation (STD) in seconds:
>
> **CIFAR-10 - ResNet56**
> | Method | Average Step | STD |
> |----------|----------|----------|
> | $l1$   | 0.0815 (s)  | 0.0177 (s)   |
> | NEXP   | 0.8415 (s)   | 0.0674 (s)   |
>
> **Imagenet - ResNet50**
> | Method | Average Step | STD |
> |----------|----------|----------|
> | $l1$   | 0.3789 (s)   | 0.1498 (s)   |
> | NEXP   | 0.4720 (s)   | 0.1336 (s)   |
>
> Additionally, the paper demonstrates that NEXP exhibits resilience to a limited number of samples, as evidenced by the significant correlations outlined in A, particularly in A.2. Consequently, NEXP can sustain consistent measurements without a significant decrease in computational efficiency as the models and tasks become more complex. Conversely, weight-based pruning methods, such as $l1$, exhibit increased computational overhead as model sizes grow, as highlighted in the aforementioned tables. This increase is most of the times caused due to their dependency on the dimensions and/or cardinality of the layer weight matrices. In contrast, NEXP does not correlate with model size and thus demonstrates prominent scalability in computational efficiency as model complexity increases.
>
> **W3.** Overall, we concur with the pruning evaluation framework articulated by Blalock, Davis, et al. [1], *"Pruning imposes a tradeoff between model efficiency and quality, with pruning increasing the former while (typically) decreasing the latter. This means that a pruning method is best characterized not by a single model it has pruned, but by a family of models corresponding to different points on the efficiency-quality curve."*. For the experimental results of object detection (lines 287-315), the assessment of the hybrid optimization space (4.2), and the evaluation of NEXP at initialization (4.3), different points on the efficiency-quality curve for all methods are sampled based on a consistent compression format with equal compression ratios. Each point on the curve is generated either based on a target ($\tau$) FLOPs compression ratio (see Algorithm 1) for object detection and hybrid optimization or on a target ($\tau$) params compression ratio (4.3, see lines 675-678). However, for the comparison on image classification, the results of all adversarial pruning methods are directly sourced from their respective publications and not replicated locally. Thus, to avoid excessive result tables, we have defined different FLOPs compression regimes of interest to ensure a fair comparison. To improve the clarity of the paper, we have extended lines 270-272 to include the following: *"The results of adversarial pruning methods in this subsection are directly obtained from the respective publications."*
>
> **W4.** The insights of most adversarial methods are discussed in Related Work, specifically in lines:
> - 113-115: L1, GAL, LAMP
> - 117-120: NISP, ThiNet, GAL, LAMP
> - 121-123: DCP
> - 131-132: HRank
>
> An outline of their motivations is also provided in lines 270-272 and C.2. We consider these sections essential as they establish the mathematical foundation for conceptualizing and differentiating between importance and expressiveness, and highlight the motivation behind NEXP (see response in W2 for the rebuttal to reviewer's pBA2 comments). However, in case of acceptance, space of the extra page will be reserved to provide further insights on the later introduced methods.
>
> **Q2.** Fig. 3 suggests that NEXP is consistently superior to l1 in compression efficiency, aligning with previous sections findings that NEXP consistantly achieves higher parameter compression ratios for given target (τ) FLOPs compared to weight-based methods. As outlined in [2], the pruned architecture, rather than inherited 'important' weights, is more crucial to the efficiency of the final model. Interestingly, hybrid optimizations can facilitate strategies that will further explore the trade-off between the quality of the pruned architecture and the inherited 'important' weights.
>
> **Q4.** A detailed description of Fig. 1 is in lines 247-251 (3.3). Appendix A provides a comprehensive discussion of the findings related to Fig.1, while Fig. 4 presents an extended version, including Expressiveness at Initialization.
>
> **Q1, Q3.** Kindly refer to the responses in Q1 and Q2.b (respectively) for the rebuttal to reviewer's ha2w comments.
>
> [1]. Blalock, Davis, et al. "What is the state of neural network pruning?." Proceedings of machine learning and systems 2 (2020): 129-146.
>
> [2]. Liu, Zhuang, et al. "Rethinking the value of network pruning." arXiv preprint arXiv:1810.05270 (2018).

---

> > ### Comment · Reviewer_9m28 · 2024-08-12
> > **Rebuttal**
> >
> > I would like to thank the authors for the rebuttal. My concerns are mostly addressed. Thus I increase my score. I would suggest that the authors include the speed evaluation of the pruning method in the updated version.

---

> ### Author Response · Authors · 2024-08-13
> **Reply to Reviewer 9m28**
>
> Thank you for taking the time and effort to address our initial rebuttal (especially at this time of year). Your insightful comments and suggestions have given us the opportunity to further clarify the intuition and effectiveness of our proposed approach, while also improving the quality of the paper. A discussion on the **Computational Scalability** of NEXP, as well as a comparison (both theoretical and experimental) against weight-based methods, has been included in the updated version of the paper.
>
> We greatly appreciate your increased score.

---

### Official Review · Reviewer_4Yxj · 2024-07-20

**Soundness:** 3
**Presentation:** 3
**Contribution:** 2
**Rating:** 4
**Confidence:** 4

**Summary:**

This paper aims to handle network pruning problem. Specifically, it proposes to use a new importance measurement, called expressiveness, to decide the pruning process. It jointly considers the model state to leverage on the proposed measurement. In addition, it can also combined with typical importance based pruning methods to improve the model efficiency.

**Strengths:**

1. Network pruning is a valuable research direction to study on, espeicailly for current large-scale era where efficiency matters a lot.
2. The proposed new metric to measure the network importance is interesting.
3.  Empirical results show the method superiority.

**Weaknesses:**

1. Adding more discussion in the conclusion part helps to improve the paper readability.
2. Since this paper proposes a new pruning metric, it is better to show more visualization and network behavior analysis to illustrate the intuition.
3. The compared baseline models are relatively old, adding more recent publications helps to support this paper.

**Questions:**

Please refer to the weaknesses section above.

---

> ### Author Rebuttal · Authors · 2024-08-06
>
> **W1.** The submitted version of the paper features a brief conclusion section to adhere to the 9-page limit of the NeurIPS format, prioritizing space for other sections. We agree that an extended discussion in the conclusion is vital for the paper's overall remarks and readability. Below, we provide the full version of the conclusions, which will be incorporated into the paper if accepted, as an additional content page will be permitted for the camera-ready version.
>
> **Conclusions and Future Directions:** *In this work, we have introduced "Neural Expressiveness" as a new criterion for estimating the contribution of a neuron based on its ability to extract features that maximally separate sub-spaces within the feature space, using the overlap of activations. We demonstrated that neural expressiveness can be approximated with limited arbitrary data and that is capable of yielding consistent estimations across the evolutionary (learning) states of a neural network. We also provided a theoretical background and mathematical conceptualization on the differentiators, along with an experimental study on the complementary nature between expressiveness and previous weight-based (importance) methods, hinting at potential future directions. Finally, after extensive experimentation, we showcased the efficacy of neural expressiveness, consistantly delivering significant compression ratios, across various scheduling, fine-tuning, and evaluation setups, with minor deviations in performance, and outperforming state-of-the-art solutions in both PaT and PaI. In our future NEXP steps, based on the intrinsic properties of expressiveness presented in this work, we aim to investigate optimization solutions for exploring the hybrid compression solution space, as well as solutions for bringing model compression closer to the initialization state, addressing questions such as, 'Is neural network training essentially about learning new knowledge from scratch, or is it about revealing the knowledge that the model already possesses?' (as quoted from Wang et al. [1]), and thus minimizing the need for extensive training iterations.*
>
> **W2.** We have included **Figure r1** in the 1-page rebuttal PDF, which has been incorporated in Subsection 3.1 (*"Weights and Activations: Importance vs Expressiveness"*) to facilitate a better understanding of the notations and intuitions in Section 3. This figure visually illustrates the introduced notations, clarifying the intuition behind the proposed method's motivation and differentiation from the intuition of existing pruning approaches. (This was also motivated by reviewer pBA2 W1's comment.)
>
> W2.a. A two-fold sensitivity analysis of NEXP is provided in Appendix A. This analysis examines the dependency of the proposed method to (i) the mini-batch data ($X$, as outlined in Alg. 1), using two input sampling strategies—random sampling and class-representative sampling via k-means—and (ii) the information state ($W_{t_i}$), specifically comparing expressiveness at initialization against expressiveness after training, when weights have converged, for various neural networks (i.e., ResNet-56, MobileNet-v2, DenseNet-40, and VGG19).
>
>
> **W3.** Thank you for your attention to detail. We have extended the paper to incorporate the following recently published pruning approaches [2, 3]. Below, we provide an outlook on the updated result tables for different target ($\tau$) theoretical speed-up ratios (#FLOPs ↓), showcasing a comparison between the newly included methods and our proposed approach (NEXP).
>
> **CIFAR-10 - ResNet-56**
> | Method | Base (%) | ∆ (%) | #Params ↓ | |
> |----------|----------|----------|----------|----------|
> | NUCLEAR [2]  | 93.59 |   -0.07 | 1.16$\times$ | 1.45$\times$ |
> | NEXP (Ours)  | 93.36  |   +0.05 | **1.69$\times$** | 1.53$\times$ |
> |-------------|----------|-------|---------------|---------------|
> | NUCLEAR [2]  | 93.59 |   -0.30 | 1.81$\times$ | 1.76$\times$ |
> | DTP [3]  | 93.36 | +0.12 | 2.01$\times$ | 1.99$\times$ |
> | NEXP (Ours) | 93.36 | -0.41 | **2.87$\times$** | 2.11$\times$ |
> |-------------|----------|-------|---------------|---------------|
> | NEXP (Ours) | 93.36 | -1.58 | **4.3$\times$** | 2.50$\times$ |
> | NUCLEAR [2]  | 93.59 |   - 1.94 | 2.83$\times$ | 2.77$\times$ |
> | DTP [3]  | 93.36 | -0.90 | 3.39$\times$ | 3.59$\times$ |
> |-------------|----------|-------|---------------|---------------|
> | NEXP (Ours) | 93.36 | -5.12 | **21.5$\times$** | 5.00$\times$ |
> | DTP [3]  | 93.36 | -3.52 | 10.41$\times$ | 11.41$\times$ |
> | DTP [3]  | 93.36 | -7.18 | 20.79$\times$ | 19.31$\times$ |
>
> **CIFAR-10 - DenseNet40**
> | Method | Base (%) | ∆ (%) | #Params ↓ | #FLOPs ↓|
> |----------|----------|----------|----------|----------|
> | FROBENIUS [2]  | 94.82 |   -0.13 | 1.76$\times$ | 1.67$\times$ |
> | *(new line)* NEXP (Ours)  | 94.64  |  -0.17  | **1.91$\times$** | 1.70$\times$ |
>
> **CIFAR-10 - MobileNet-v2**
> | Method | Base (%) | ∆ (%) | #Params ↓ | #FLOPs ↓|
> |----------|----------|----------|----------|----------|
> | NEXP (Ours)  | 94.32  |  **+0.13**  | 2.25$\times$ | 2.11$\times$ |
> | DTP [3]  | 93.70 |  -1.77  | 2.50$\times$ | - |
>
> In general, the selection of reported adversarial compression results in this study was constrained by the frequent absence of size compression ratio reporting (i.e., parameter reduction) in many pruning studies.
>
> [1]. Huan Wang, Can Qin, Yue Bai, Yulun Zhang, and Yun Fu. Recent advances on neural network pruning at initialization. In IJCAI, 2022.
>
> [2]. Sun, Xinglong, and Humphrey Shi. "Towards Better Structured Pruning Saliency by Reorganizing Convolution." Proceedings of the IEEE/CVF Winter Conference on Applications of Computer Vision. 2024.
>
> [3]. Li, Yunqiang, et al. "Differentiable transportation pruning." Proceedings of the IEEE/CVF International Conference on Computer Vision. 2023.

---

### Author Rebuttal · Authors · 2024-08-07

We would like to thank the reviewers for their thoughtful feedback and their comprehensive suggestions.
*(Each review has been addressed separately)*

---

### Decision · Program_Chairs · 2024-09-25

**Decision:**

Reject

**Comment:**

The reviewers are not fully convinced by the paper. Although the authors could sort out some issues in the rebuttal, the reviewers ratings are 2 x 4 and 2 x 5, making this paper a clear borderline case. After carefully reading the paper and the discussions with the reviewers, I came to the conclusion that the paper is below the acceptance threshold. Although the paper has some merits and is certainly of interested to the NeurIPS community, the strength of the contribution is limited as mentioned by reviewer ha2w.
Therefore, the paper can not be accepted at NeurIPS.